# Duff burning from wildfires in a moist region: different impacts on PM$_{2.5}$ and Ozone

Aoxing Zhang[1], Yongqiang Liu[1], Scott Goodrick[1], Marcus D Williams[1]

[1]Center for Forest Disturbance Science, US Forest Service, 320 Green St., Athens, 30602, United States

*Correspondence to*: Yongqiang Liu (yongqiang.liu@usda.gov)

**Abstract.** Wildfires can significantly impact air quality and human health. However, little is known about how different fuel bed components contribute to these impacts. This study investigates the air quality impacts of duff and peat consumption during wildfires in southeastern United States, with a focus on the differing contributions of fine particulate matter less than 2.5 μm in size (PM$_{2.5}$) and ozone (O$_3$) to air quality episodes associated with the four largest wildfire events in the region during

this century. The emissions of duff burning were estimated based on a field measurement of a 2016 southern Appalachian fire. The emissions from the burning of other fuels were obtained from the Fire INventory from NCAR (FINN). The air quality impacts were simulated using a 3-D regional air quality model. The results show the duff burning emitted PM$_{2.5}$ comparable to the burning of the above-ground fuels. The simulated surface PM$_{2.5}$ concentrations due to duff burning increased by 61.3% locally over a region approximately 300 km within the fire site and by 21.3% and 29.7% in the remote metro Atlanta and

Charlotte during the 2016 southern Appalachian fires, and by 131.9% locally and by 17.7% and 24.8% in the remote metro Orlando and Miami during the 2007 Okefenokee fire. However, the simulated ozone impacts from the duff burning were negligible due to the small duff emission factors of ozone precursors such as NO$_x$. This study suggests the need to improve the modeling of PM$_{2.5}$ and the air quality, human health, and climate impacts of wildfires in moist ecosystems by including duff burning in global fire emission inventories.

## 1 Introduction

Wildfires, caused by natural factors or human activities, have a fundamental impact on air quality, human health, and climate. Wildfires contribute up to 40% organic carbon (OC) emissions in Europe, 42% in Asia, 64% globally, and dominate the regional particular matters (PM) concentrations over the major fire regions in Africa and South America (Granier et al., 2011; Diehl et al., 2012). Fires contribute 26.9% of total volatile organic compounds (VOC) emissions and 27.5% of PM emissions

in the U.S. according to the 2014 US Environmental Protection Agency (EPA) National Emissions Inventory (NEI) (USEPA, 2017). Wildfires are large sources of atmospheric aerosols (Crutzen and Andreae, 1990; Bond et al., 2005; Bowman et al., 2009; Brey and Fischer, 2016), contributing 30% of the aerosol optical thickness (AOT) in Europe (Hodzic et al., 2007), more than 80% in the Amazon area during the fire season (Reddington et al., 2019), and 10% globally (Tosca et al., 2013). In the contiguous US during 2008 - 2012, fires contribute 11% of the total PM$_{2.5}$ concentrations (Wilkins et al., 2018).

Wildfires emit tracer gases including ozone precursors and therefore contribute to tropospheric ozone, a critical air pollution compound that adversely impacts human health (McKee, 1993). Ozone production has been detected in fire plumes (Goode et al., 2000; Jaffe et al., 2008). Wildfires accounts for 3.5% of global tropospheric ozone production, though ozone production rates of individual fires vary with location, time, fuel type, combustion efficiency, meteorology, and local pre-existing atmospheric composition, etc. (Alvarado et al., 2010; Jaffe and Wigder, 2012). In the United States, when fires are present, 14% of simulated maximum daily 8-h average ozone concentrations surpassed 70 ppb (Wilkins et al., 2018), which is the standard from EPA.

High-severity fire events have frequently impacted metropolitan regions. For example, the smoke from the 2013 Rim Fire and wildfires during 2017 and 2018 in California, US was transported long-range and affected large urban areas (Liu et al., 2016; Navarro et al., 2016; Mass and Ovens, 2019; Brown et al., 2020). The smoke from the 2009 Attica forest fires decreased the surface solar irradiance levels by 70% in Athens, Greece (Amiridis et al., 2012). The 2017 Italian Alps fire had a significant impact on metro Torino, Italy (Bo et al., 2020). Similar fire events impacted urban air quality in other regions around the world (Shaposhnikov et al., 2014; Mallia et al., 2015; He et al., 2016; Cuchiara et al., 2017). In many regions around the world, including the U.S., wildfires have an increasing trend during recent decades in both the number and the area of total large fires (Dennison et al., 2014; Barbero et al., 2015). In addition, weather with high fire potential has appeared more frequently. (Yang et al., 2011; Jolly et al., 2015; Abatzoglou and Williams, 2016), leading to an increasing concern on their adverse impact on air quality (Singh et al., 2012; Goodrick et al., 2013; Liu et al., 2014; Zhang and Wang, 2016).

Negative impacts of wildfires on human health are devastating when smoke plumes are transported to populated metropolitan areas (Kunzli et al., 2006). Epidemiological studies have revealed fire emissions' contribution to $PM_{2.5}$ oxidative potential, which is related to respiratory and cardiovascular diseases (Verma et al., 2014; Yang et al., 2016; Fang et al., 2016). During the fire events in northwestern U.S. during August-September, a regional mortality of 183 due to $PM_{2.5}$ exposure was estimated, in which 95% was contributed by fire emissions (Zou et al., 2019). Based on the U.S. respiratory hospital admissions and additional premature deaths during and after fire events, the economic loss is \$11 – 20 billion due to short term exposures, and \$76 – 130 billion due to long-term exposures (Fann et al., 2018).

Several datasets and three-dimensional atmospheric models have been used to understand the amount, transport, and physical and chemical processes of fire emissions (Liu et al., 2020; Pan et al., 2020). Some widely used global fire emission inventories include the Global Fire Emission Dataset (GFED) (Randerson et al., 2012; Giglio et al., 2013; Van Der Werf et al., 2017), Fire INventory from NCAR (FINN) (Wiedinmyer et al., 2006; Wiedinmyer et al., 2011), Global Fire Assimilation System (GFAS) (Kaiser et al., 2009; Kaiser et al., 2012), Fire Energetics and Emissions Research (FEER) (Ellison et al., 2014), and Quick Fire Emissions Dataset (QFED) (Darmenov and da Silva, 2013). Global atmospheric models such as the Community Earth System

Model (CESM) were used to study wildfire smoke transport and interactions with land and atmosphere (Jiang et al., 2020; Zhang et al., 2020; Zou et al., 2020) and the GEOS-Chem model was used to evaluate the wildfire contribution to atmospheric chemistry (Lu et al., 2016). Regional air quality models, such as the Weather Research and Forecasting model with Chemistry (WRF-Chem) and the Community Multiscale Air Quality (CMAQ) model have higher spatial resolutions, thus have advantages when simulating fire smoke aging and regional plume transport (Jaffe et al., 2008; Lu and Sokolik, 2017; San Jose et al., 2017; Wilkins et al., 2018; Guan et al., 2020).

Emissions from duff burns are an important contributor to the global carbon cycle. Duff typically represents the detritus or dead plant organic materials fallen at the top layer of soil. Besides duff, peat is another burnable organic soil that typically represents the fermentation below the duff layer (Frandsen, 1987). "Organic soil" is often used to represent soil formed by plant and animal decomposition, including peat and duff. Duff, peat and organic soil were sometimes used interchangeably, and we focus on duff in this study. Temperate and boreal duff layers are well distributed in forests and swamps in North America, Europe and Asia (Wieder et al., 2006). Compared to the burning of above-ground fuel, duff burning can have a similar or larger amount of carbon emission, enlarging the regional and global effect from wildfires (Ballhorn et al., 2009; Reddy et al., 2015). Ground-based studies have been conducted to estimate the carbon loss from temperate duff flaming or smouldering. Davies et al. (2013) surveyed the peatland smouldering in the Scottish Highlands, UK and estimated a $17.5 \pm 2.0$ cm burned depth of below-ground fuel and $9.6 \pm 1.5$ kg m$^{-2}$ carbon loss due to smouldering. In North Carolina, US, the 1985 Pocosin Lakes fire resulted in a carbon flux of 0.2 - 11 kg m$^{-2}$, that varies with burned depth, vegetation type and burning severity (Poulter et al., 2006). Assuming 50% of the duff mass is carbon (Watts, 2013), this fuel loading results in a carbon loss of approximately 1.6 kg m$^{-2}$. Watts (2013) estimated 4.18 kg m$^{-2}$ carbon release from the wetland combustion in the Big Cypress National Preserve in southern Florida, US. Duff and peat are a major reservoir of wetland carbon and contribute 3% of global land cover (Gorham, 1991; Yu et al., 2010). The burning properties and emission factors of the below-ground organic soils, including duff and peat, are similar (Raaflaub and Valeo, 2009; Urbanski, 2014). The air quality impacts from peatland burning have also been evaluated in tropical peatlands in Indonesia (Page et al., 2002; Kiely et al., 2020).

However, the air quality impacts of emissions from duff fires are still not well understood over some regions (Page et al., 2002; Hu et al., 2018). One of the reasons is the lack of the fire emission data (Ward et al., 2012), which is a large uncertainty source for simulations of the fire impacts on air quality. Satellite remote sensing is a very useful tool to obtain fire emissions with detailed global and regional coverage. The organic soil burning over tropical peatlands has been considered in GFED (Randerson et al., 2012; Giglio et al., 2013; Van Der Werf et al., 2017). Indonesia peat fire emissions are also updated and evaluated in FINN (Kiely et al., 2019). However, compared to above-ground fuel burning, duff emissions are not documented enough by satellite-based global fire emission datasets in forest ecosystems, partially due to the presence of overhead canopies. Another reason is that duff burning usually occurs during the smouldering phase because of the relatively high soil moisture (Ottmar, 2014). Smouldering often lasts over a long time. Also, the emissions do not rise to high elevations due to low heat

release. Thus, the emissions, especially particles, have little impact on regional air quality in populated areas far from the source region.


Similar to many world regions (e.g. tropical forests in southeast Asia (Page et al., 2002) and temperate forests in the Great British Isles (Davies et al., 2013)), the southeastern US is a duff-rich region because of the high humidity and large forest coverage (Zhu and Evans, 1994; Gaffen and Ross, 1999). The warm and moist climate makes vegetation growth and falling leaves and branches decompose fast and therefore accumulate as deep duff, especially in the southern Appalachians (Ottmar

and Andreu, 2007) and the Okefenokee swamp (Watts and Kobziar, 2012), which are located in the northern and southern portions of this region, respectively. This region has some unique features among all US regions in the contributions to the carbon cycling and regional air pollution. On one hand, most fires in the southeastern US are prescribed (planned) and conducted in weather where duff consumption is minimized (Waldrop and Goodrick, 2012). Thus, duff burn may be only a small contributor to total pollutant emissions in this region. On the other hand,  there are large wildfires that occur under

drought conditions and are close to populated areas, although the frequency and severity are usually small relative to those in the western US (Goodrick et al., 2013). Wildfires in the southeastern US usually occur in spring before the summertime rain season starts. Sometimes wildfires can occur in other seasons under drought conditions such as the southern Appalachians fire in fall 2016.  As described above, duff burning usually occurs during smouldering phase, however, this situation is changing with more frequent occurrences of droughts, which increases the flammability of the duff layer (Hille and Stephens, 2005).

Duff burn during flaming phase of the 2016 Rough Ridge Fire in the southern Appalachian, which occurred  during a prolonged severe drought (Park Williams et al., 2017), was reported by fire managers and the related fuel consumption was measured (Zhao et al., 2019). The measured duff layer burned by the fire was 4.6 cm deep with 31.5 Mg ha$^{-1}$ (3.15 kg m$^{-2}$) fuel loading, estimated to account for approximately 60% of total PM2.5 emitted from the fire. The simulations including duff emissions conducted by Zhao et al. (2019) indicated that the duff burn was a major contributor to the air pollutions in the nearby metro

Atlanta. In contrast, a model simulation study on all major 2016 southern Appalachian fires that excluded duff burning resulted in an underestimation of PM$_{2.5}$ during the fire events (Guan et al., 2020).

The Okefenokee swamp experienced fires during the dry years of 2007, 2011, and 2017, each with much larger burned area than the total burned area from the 2016 southern Appalachian fires. The burning of the duff layer was reported during all

three fire events in the Okefenokee region (The 2007 Big Turnaround Fire: https://www.fws.gov/fire/downloads/fire_updates/BigTurnaround.FINAL.pdf, the 2011 Honey Prairie Fire: https://www.wunderground.com/blog/weatherhistorian/the-great-okefenokee-swamp-fire-of-2011.html; the 2017 West Mims Fire: https://gatrees.org/wp-content/uploads/2020/02/Wildfire-Damage-Assessment-for-the-West-Mims-Fire.pdf). However, it is not clear how much the duff burning from these fire events contributed to air pollutions in the populated areas.


The literature is still not conclusive on the differing impacts of duff burning on various air pollutants. The emission factors of duff are different from those of above-ground fuels (Yokelson et al., 2013; Urbanski, 2014; Hu et al., 2018; Kiely et al., 2019). For example, the temperate forest duff emission factor of nitrogen oxides ($NO_x$) is 0.67 g/kg (Yokelson et al., 2013), more than 50% smaller than the conifer forest emission factors. However, the temperate forest duff emission factor of $PM_{2.5}$ is 50

$\pm$ 16 g/kg (Geron and Hays, 2013), which is more than twice of the $PM_{2.5}$ emission factors from conifer forests (13 - 23 g/kg) (Yokelson et al., 2013; Urbanski, 2014). Because $NO_x$ is a major precursor of ozone formation, these different emission factors potentially lead to a stronger $PM_{2.5}$ impact than ozone impact for duff burning.

The goal of this study is to investigate the contributions of duff burning from the largest wildfires this century in the
southeastern US to regional air pollutions and the differences between $PM_{2.5}$ and ozone. The simulations of regional smoke transport were conducted based on the duff measurements from the Rough Ridge Fire (Zhao et al. 2019) and the global fire emission dataset from FINNv1.5. The simulated concentrations of air pollutants were compared with those from observations, between burns with and without duff, and between PM2.5 and ozone. The results are expected to provide important implications for the needs in improving global fire emission inventories and understanding the contributions of duff and peat
burnings in other world regions to regional air pollution.

## 2 Methods

### 2.1 Study region

The study region is the southeastern US, which comprises the states of Florida, Alabama, Georgia, South Carolina, North Carolina, Tennessee, Mississippi, and Louisiana. This region is dominated by a humid subtropical climate (Belda et al., 2014).
The summers are typically long with high temperature and humidity, contributed by the water vapor transport from Bermuda High (Li et al., 2011). The winters are typically dry in peninsular Florida, but relatively wet in the mid-south, such as Tennessee and the northern Georgia and Alabama (Gaffen and Ross, 1999). The ecozones in the southeastern US include broadleaf forest over the Appalachian region in the west of North Carolina and South Carolina, and the north of Georgia, and mixed forest in the other regions including most Georgia and Florida (Bachelet et al., 2001; Blood et al., 2016). Hardwood and pine are major
above-ground fuels in the southeastern US (Ottmar and Andreu, 2007). Because of the sufficient light and the regularly high humidity the duff layer is accumulated in the southeastern US and contributes as the potential below-ground fuel, especially in wildlife refuges or regions where deciduous trees are widely distributed with lack of prescribed burn removal.

The wildfire cases investigated in this study occurred in three areas. The first is the southern Appalachian Mountains in the
northern part of the southeastern US. This area is located in the boundaries of Georgia, North Carolina, South Carolina, Tennessee, Virginia, West Virginia, and Kentucky. The Southern Appalachian region is deciduous forests dominated, with small proportions of evergreen forests and mixed forests. The main forest type is the hardwood oak forest (Southern

Appalachian Man and the Biosphere, 1996). The second area is the Okefenokee Swamp located across the Georgia-Florida border. The 438,000 acres swamp is mainly covered by the Okefenokee National Wildlife Refuge. Cypress forests and scrub-shrub vegetation are the major vegetation types over the Okefenokee region, and the wetland is covered by a duff layer with a depth of up to 4.6 m (Url: https://www.fws.gov/refuge/okefenokee/, last access: November 17, 2020). The Okefenokee region is sensitive to rainfall. Under drought conditions, the region is vulnerable to wildfire. The third area is coastal eastern North Carolina. The ground forest fuels are rich of organic decomposition due to the moist and warm climate. There are many populated metros nearby in the west including Raleigh-Durham-Chapel Hill triangle. Smoke from fires in this area can be transported to affect these metros rapidly under easterly winds.

## 2.2 Fire cases

In this study, we investigated six wildfire cases (Table 1), and the map of the fire cases over the studies regions shown in Fig. 1 and Fig. S2. The first case included 10 large fires from mid-October to December 2016 in the southern Appalachian mountains during an extreme drought (Konrad and Knox, 2017; Park Williams et al., 2017). The fires burned 91,191 acres of forest, caused losses of 14 lives and massive property loss (McDowell et al., 2017; Pouliot et al., 2017). The largest fires were the Rough Ridge Fire (34.88° N, 84.63° W, ignited on October 16, 27,610 acres burned), the Rock Mountain Fire (34.98° N, 83.52° W, ignited on November 9, 25,224 acres burned), and the Tellico Fire (35.28° N, 83.58° W, ignited on November 3, 14,172 acres burned). We denote this case as App16.

The next three cases occurred in Okefenokee in 2007, 2011, and 2017, respectively. We denote them as Oke07, Oke11, and Oke17. The 2007 Okefenokee mega wildfire was ignited in the Okefenokee Wildlife Refuge (30.67° N, 82.45° W) on April 16, and had burned more than 500,000 acres until late June (Fire Behavior Assessment Team, 2007). Protracted drought led to low water levels in the Okefenokee swamp and provided the condition of burning in a mix of shrub scrub, wetland prairies, duff, cypress and long-leaf pine forests. This fire remains the largest wildfire in the history of Georgia and Florida (Url: https://www.fws.gov/fire/downloads/fire_updates/BigTurnaround.FINAL.pdf, last access: October 29, 2020).

The 2011 Honey Prairie Fire was ignited on April 30 under a severe drought, during which the Okefenokee Swamp water level was lower than that during the 2007 mega-fire (https://www.fws.gov/refuges/news/HoneyPrairieFire_05112011.html, last access: December 3, 2020). 147,065 acres were burned (Finco et al., 2012). The 2017 West Mims Fire was ignited on April 6 under an extreme drought and developed fast in early May (http://www.gatrees.net/forest-management/forest-health/alerts-and-updates/Wildfire%20Damage%20Assessment%20for%20the%20West%20Mims%20Fire.pdf, last access: December 3, 2020). 166,737 acres were burned. Historical fire records from 19[th] century revealed the strong connection between drought and Okefenokee fires, leading to a 'Okefenokee drought-fire cycle' (https://www.frames.gov/catalog/34075, last access: November 5, 2021). Although the Okefenokee fires from 2007 to 2017 was more frequent than the historical mean, more information on the fire cycle change is needed.

The other two fire cases occured in the coastal southeastern US. The 2008 Evans Road Fire was ignited to the south of the Pocosin Lakes National Wildlife Refuge, North Carolina on June 1, 2008 by lightning. 41,060 acres were burned (https://files.nc.gov/ncdeq/Air%20Quality/monitor/specialstudies/exceptionalevents/2008/Exceptional%20Event%20Evans %20Road%20Fire.pdf, last access: October 27, 2021). The 2011 Pains Bay Fire was ignited in the FWS Alligator River National Wildlife Refuge in the coastal North Carolina, 29,400 acres were burned (https://www.geobabble.org/~hnw/first/EWSNews/EWS_Fire_PainsBay_2011-0601.pdf, last access: October 27, 2021). Significant organic/ground fuels were burned during the two coastal fires, causing subsequent air quality impacts and health impacts (Rappold et al., 2011; Tinling et al., 2016). We denote these two fire cases as ER08 and PB11, respectively.

## 2.3 Model simulations

### 2.3.1 Model

The model components and implementation procedure used for simulations are illustrated in Fig. 2. We used WRF-Chem version 3.9.1 (Grell et al., 2005; Fast et al., 2006; Powers et al., 2017) to simulate the aerosol, gas transport, and atmospheric chemistry over the southeastern US. The model has coupled gas-phase atmospheric chemistry (Wang et al., 2015; Zhang et al., 2016), aerosol optical properties (Barnard et al., 2010), and the new Thompson graupel microphysics scheme (Thompson et al., 2008). The radiation scheme is the Rapid Radiative Transfer Method for Global Climate Models (GCMs) (RRTMG) (Iacono et al., 2008; Mlawer et al., 1997). The kinetic preprocessor (KPP) library was used for chemical reactions (Damian et al., 2002; Sandu et al., 2003; Sandu and Sander, 2006). The 1° x 1° meteorological data from National Centers for Environmental Prediction (NCEP) FNL (final) Operational Model Global Tropospheric Analyses (FNL NCEP, 2000) was used as the meteorological initial and boundary conditions for the simulations.

The Model for Ozone and Related chemical Tracers (MOZART) (Emmons et al., 2010) was used as the WRF chemistry module, coupled with Georgia Tech/Goddard Global Ozone Chemistry Aerosol Radiation and Transport (GOCART) aerosol scheme (Chin et al., 2002). Madronich F-TUV photolysis scheme was applied, with a time step of 15 minutes (Madronich, 1987). The time step was 3 minutes for chemistry. Gas and aerosol dry deposition, aerosol wet scavenging, vertical mixing, subgrid convective transport, and subgrid aqueous chemistry (Peckham et al., 2018) were included in the model simulations. The global simulation result from MOZART (Pfister et al., 2011b) was used as the chemical initial and boundary conditions of the simulation in this study. The ozone initial and boundary conditions from MOZART were also scaled by comparing the mean surface ozone concentration over the simulation domain with the US EPA Air Quality System (AQS) observations (https://www.epa.gov/outdoor-air-quality-data, last access: October 22, 2020).

### 2.3.2 Simulation domains

The simulation domain for App16 was from 30.4° N to 37.5° N and from 88.3° W to 77.7° W, with a spatial resolution of 12

km. This domain included the major burning sites and the downwind nearby large cities including Atlanta (33.75° N, 84.39° W) and Charlotte (35.23° N, 80.84° W). The simulation period was November 7 - 22, 2016. The daily trend of FINN organic carbon (OC) emissions over the fire region is shown in Fig. S1, indicating that the simulation period cases contained the most severe burning that occurred during the fire case.

The simulation domain for the three Okefenokee cases was from 23.9° N to 37.0° N and from 92.6° W to 72.4° W, with a

spatial resolution of 12 km (Fig. 1). The Okefenokee Wildlife Refuge was located at the center of this domain. Nearby cities and the ocean were included to evaluate the smoke transport to urban and remote areas. The simulation periods were May 6-30, 2007, May 4-15, 2011, and April 19 - May 13, 2017, respectively. The simulation domain for the PB11 cases is the same as the Oke11 case, and the simulation domain for the ER08 case was from 31.7° N to 38.1° N and from 84.9° W to 74.1° W.

### 2.3.3 Simulations and evaluations

For each fire case, we conducted three simulations to evaluate the air quality impacts from fires and duff burning (Table 1). (1) sim_nofire: no fire emissions; (2) sim_FINN: FINNv1.5 fire emission dataset was used as the fire emission input, but duff burning was not included in this dataset; (3) sim_FINN+duff: same as sim_FINN but with duff burning emissions. We used the differences in the results between sim_FINN and sim_nofire to represent the impacts from fire, and the differences between sim_FINN+duff and sim_FINN to represent the impacts from duff burning.


We evaluated the model performances in simulating air pollutant concentrations by comparing them with the EPA Air Quality System (AQS) in-situ hourly observations for $PM_{2.5}$ and ozone (https://www.epa.gov/outdoor-air-quality-data, last access: October 22, 2020). Starting in 2008, EPA included the Federal Reference Methods (FRM) or Federal Equivalent Methods (FEM) for the particulate measurement as a systematic framework, which provides standard methodologies and procedures

for measuring and analyzing PM (Noble et al., 2001). During Oke07, the FRM/FEM was not spread out in the $PM_{2.5}$ measurement system from EPA AQS. For the consistency of all the fire cases, both FRM/FEM and non-FRM/FRM datasets were used for comparison. 56 $PM_{2.5}$ observation sites and 53 ozone observation sites are included in the evaluation for App16, 76 $PM_{2.5}$ observation sites and 225 ozone observation sites for Oke07, 112 $PM_{2.5}$ observation sites and 215 ozone observation sites for Oke11 and PB11, 120 $PM_{2.5}$ observation sites and 208 ozone observation sites for Oke17, 38 $PM_{2.5}$ observation sites

and 101 ozone observation sites for ER08.

The day-time surface ozone concentrations, calculated by averaging surface ozone concentrations from local time 10 am to 6 pm, were evaluated between the baseline simulations (sim_FINN) and the observations. In the model evaluation and the

following result analysis, the surface concentrations in the simulation are defined as the concentrations at the bottom layer in
the model, which was also the layer where the surface emission input was added in.

## 2.4 Emission data

### 2.4.1 Fire emissions of above-ground fuels

The fire emissions from FINNv1.5 were implemented into WRF-Chem by Pfister et al. (2011a), which contains the daily
burned area and emissions of an amount of gas and aerosol species with a spatial resolution of 1 km (Wiedinmyer et al., 2011).
No a-priori diurnal cycle of the fire emission was applied in the WRF-Chem model, and the hourly fire emission applied in
the WRF-Chem simulations was the hourly emission converted from the daily fire cases from FINN, assuming that fire at each
observed fire hotspot lasted for one day. The plume rise calculation of the fire emission using a 1-D time-dependent dynamic
cloud model was called every 30 minutes (Freitas et al., 2007; Grell et al., 2011). The high-resolution in both space and time
with the FINN fire data is a valuable feature for this study.

The biogenic emissions from the Model of Emissions of Gases and Aerosols from Nature (MEGAN) (Guenther, 2006;
Sakulyanontvittaya et al., 2008) were used as the WRF-Chem input, from which monthly biogenic emissions with a spatial
resolution of approximately 1 km were derived. The dust, dimethylsulfide (DMS) and sea salt emissions from GOCART were
included in the model (Ginoux et al., 2001; Chin et al., 2002). For the model anthropogenic emission input, we used the NEI
2014v2 hourly anthropogenic emission dataset for the US, based on the criteria pollutant emissions from the 2014 EPA
platform (USEPA, 2018b) implemented for the National Air Toxics Assessment (USEPA, 2018a). During the simulation, the
meteorological field was nudged towards the 1° x 1° NCEP FNL reanalyses (FNL NCEP, 2000) every 6 hours, using the WRF
Four-Dimensional Data Assimilation (FDDA) method (Stauffer and Seaman, 1990).

### 2.4.2 Fire emissions of duff

Current major global fire emission inventories, such as GFED and FINN (Wiedinmyer et al., 2011; Giglio et al., 2013; Van
Der Werf et al., 2017), do not include enough duff and peat emissions. The fuel loading in FINN is based on the regional
average from Global Wildland Fire Emission Model (GWEM) (Hoelzemann et al., 2004). Total fuel loading of each grid is
assigned with one of specific land cover classifications. Litter is included in GWEM, but peat and duff are not. In contrast,
duff is explicitly included in GFED. GFED4s assumes the $PM_{2.5}$ emission factor of 9.1 g/kg from duff and peat burning
based on Andreae and Merlet (2001), which is smaller than the above-ground fuel emission factors, and significantly smaller
than the recent field and experiment results (Yokelson et al., 2013; Urbanski, 2014). Andreae (2019) updated the $PM_{2.5}$
emission factor of peat burning to 18.9 g/kg, which is larger than the above-ground fuel emission factors, but the latest fire
emission inventory has not been updated accordingly yet.

The amount of duff burned during the fire cases investigated in this study was estimated based on the measurements from Zhao et al. (2019). During the 2016 Rough Ridge Fire, 4.6 cm of a duff layer was burned within one day, which accounted for more than 90% of the total duff. The duff burning was estimated to have contributed 60% of the total $PM_{2.5}$ emission. To our best knowledge, this measurement is the only duff burned depth measurement during the flaming phase in the temperate region. In previous studies, duff burning in the smoldering phase was evaluated. For example, the duff smoldering depth in North

Carolina peat fires were measured from 0.5 cm to 10 cm (Wilbur and Christensen, 1983; Poulter et al., 2006), and Watts (2013) estimated $8.9 \pm 5.2$ cm duff burn depth during the smoldering in cypress swamps in Florida. Light detection and ranging (LiDAR) instruments detected an approximately 47 cm soil elevation loss during the 2011 Lateral West fire in a swamp in Virginia (Reddy et al., 2015). Because smoldering occurs at a low temperature in the long-term (months to years) (Rein and Belcher, 2013), which just creates weak and low plumes, here we only studied the regional air quality impact from duff flaming.


    The duff emission estimation in this study is described in Fig. 3. We estimated duff emissions and added them to FINN with the following method. First, we calculated the daily duff mass burned, $M(x, y, t)$ (kg day$^{-1}$), in the burning case over the model grid box $(x, y)$ on the day $(t)$:

$$M(x, y, t) = a(x, y, t)h\rho ,  \tag{1}$$

where $a(x, y, t)$ is burned area (m$^2$), $h$ is the average duff-layer depth burned daily in the case ($h = 0.045$ m day$^{-1}$ assumed), and ρ is the density of duff, which was assumed to be 57.4 kg m$^{-2}$ m$^{-1}$ according to the measurements over the southeastern US with the vegetation type of Pine and Hardwoods (Ottmar and Andreu, 2007). The measurements from different locations showed a 21% standard error of mean duff density.

The duff emissions were then added to FINN fire emission $E(x, y, t)_{FINN+duff,s}$ (kg day$^{-1}$) for each grid box, day, and species $(s)$:

$$E(x, y, t)_{FINN+duff,s} = E(x, y, t)_{FINN,s} + M(x, y, t) * EF_s * 0.001 ,  \tag{2}$$

    where $E(x, y, t)_{FINN,s}$ is the original FINN fire emission, and $EF_s$ is the duff emission factor of the species s (g/kg).

The $PM_{2.5}$ emission factor of duff / peat burning varies noticeably among the studies across the world regions and ecosystems (Table S1). The four studies in the southeastern US obtained average values of about 50 g/ kg (a field study that made in-situ measurements of $PM_{2.5}$ emission factors from three different peat fires in coastal North Carolina, Geron and Hays, 2013), 5.5 g/ kg (a laboratory study that measured EF from peat core samples from two locations in North Carolina, Black et al. 2016), 44 g/ kg (Benner 1977), and 30 g/ kg (McMahon et al. 1980). The first two studies took soil samples from the same peat

location in eastern North Carolina, US. Due to a previous fire investigated by the first study, the sample from the second one had much less carbon but more ash. This was a major reason for the much lower PM2.5 emission factor proposed by the

authors. For this reason, we did not consider the value from the second study when we specified the emission factor value for our study. The value from the first study was used as the US temperate duff burning emission factor in the review paper by Urbanski (2014). It was also used in our study because it is likely to better represent the burning on the vegetation type in the

southeastern US. Previous studies of measuring the $PM_{2.5}$ emission factors of duff/peat/organic soil burning are summarized in Table S1.

As described in the introduction section, the differences in emission factors between duff and above-ground fuels suggest different impacts of duff burning on $PM_{2.5}$ and ozone (Table 2). The emission factor of $PM_{2.5}$ from duff burning used in this

study (50 g kg$^{-1}$) is more than 3 times that from forest burning (13 g kg$^{-1}$). However, the emission factors of $NO_x$ from duff burning (0.559 g kg$^{-1}$ for NO and 0.176 g kg$^{-1}$ for $NO_2$) are less than 25% of those from forest fire (0.34 g kg$^{-1}$ for NO and 2.7 g kg$^{-1}$ for $NO_2$). Although the $NO_x$ emission factors vary from different locations and ecosystems, the gap of $NO_x$ emission factors from duff and the above-ground fuel was shown in different previous studies, summarized in Table S3 (Clements and McMahon, 1980; McMeeking et al., 2009; Burling et al., 2010; Selimovic et al., 2018). Except for $NO_x$ and $PM_{2.5}$, a set of

major fire emission compounds were added to duff emissions, as reported by Yokelson et al. (2013). The duff emission species considered in this study is summarized in Table 1 and Table S2.

For App16, where fires had been absent for decades before 2016 in many fire sites, emissions from duff burning were calculated using the measured depth of duff burn at the Rough Ridge Fire site. The situation was the same for Oke07. However,

some areas of Oke11 were overlapped with those of Oke07, while some burned areas of Oke17 overlapped with those of Oke07 and/or Oke11. From the FINN emission dataset, 87% of the burned area in Oke11 was burned by Oke07, and 79% of Oke17 was burned by the 2007 and 2011 fires. We assumed a duff layer recovery rate of 1 mm/year based on previous studies (Ovenden, 1990; Frolking et al., 2001; Borren et al., 2004; Milner et al., 2020). Only a fraction of the measured burned duff depth for the Rough Ridge Fire (Zhao et al., 2019) was assumed for the reburned areas. For example, if the burning during

Oke11 was also burned in 2007, only 4 mm of the duff layer was assumed to be burned and the related duff emissions were added to the 2011 sim_FINN+duff run. For the other fire cases that the burning region had not been burnt in the previous decade, a duff flaming rate of 4.6 cm/day was applied. The case mean duff flaming rate and the corresponding fuel loading is summarized in Table S4.

### 2.5 Sensitivity experiments

Many fire inventories using satellite-based models often underestimate fire emissions for a variety of reasons (clouds, small burned areas, timing, etc.) (Wiedinmyer et al., 2011) and a-priori emissions are normally scaled up to improve model - measurement agreement (Ward et al., 2012). We found that the burned area in all four wildfire cases from FINN was approximately 50% less than the burned area summarized in MTBS (Eidenshink et al., 2007), which is obtained based on vegetation changes before and after a fire rather than hot spots. The FINN emissions were also lower approximately at this

rate than the calculated emissions from based on the measured above-ground fuel consumption by the Rough Ridge Fire in northern Georgia on November 10 and 14, 2016 (Zhao et al. 2019). This FINN emission underestimate would lead to uncertainty in quantitatively estimating the contribution relative to the above-ground fuel consumption. To roughly assess the uncertainty, we did a sensitivity experiment by doubling FINN emissions for the Oke07 case (Exp_FINN, Table 1).

As described above, there are large variations in $PM_{2.5}$ and $NO_x$ emission factors. There were not enough duff measurements for the fire cases we investigated, and the duff emission factors between smouldering and flaming were also not well investigated. To evaluate the uncertainty of our simulation results due to high spatial variation of the duff layer depth, we conducted week-long sensitivity runs for App16 and Oke07 with changes of the duff burning rates by ±30% (Exp_duff, Table 1). In addition, another set of sensitivity runs was conducted for App16 and Oke07 by doubling the duff emission of $NO_x$ to

evaluate the uncertainties of the ozone effect due to the $NO_x$ emission factors (Exp_2x_duff_NOx, Table 1).

## 3 Results

### 3.1 Comparisons between simulations and observations

Here we define the fire influence based on the $PM_{2.5}$ impact from fire. If the $PM_{2.5}$ difference between sim_nofire and sim_FINN is less than 1 µg m$^{-3}$ over a specific region (and time), then this region (and time) is not influenced by fire smoke.

This value is near the low end of the thresholds often used to assess the smoke impacts (Munoz-Alpizar et al. 2017, Matz et al. 2020). For both sim_nofire and sim_FINN, the simulated $PM_{2.5}$ concentrations agree with the observations over the areas not influenced by fire events (Fig. S3). However, the baseline simulation (sim_FINN) underestimates $PM_{2.5}$ concentrations over the fire-impacted areas, shown in Fig. 4. For example, in the App16 areas (34.5° N to 36° N, 82° W to 84° W), the model underestimates $PM_{2.5}$ by 56.6% for sim_nofire and by 29.2 % for sim_FINN. For Oke07 and Oke11, the massive plume

simulated by the model is transported to a large area of Georgia. The model underestimates $PM_{2.5}$ in Georgia by 56.2 % in 2007 and 49.0% in 2011 for sim_nofire, and 47.5 % in 2007 and 39.5% in 2011 for sim_FINN. The simulated smoke from Oke17 disperses more widely in space than that from Oke11, so the intensity of the mean fire impact is minor. The comparisons of time series comparison show similar results, that is, sim_FINN underestimated $PM_{2.5}$ surface concentrations (Fig. 4). Figure 4 also shows a $PM_{2.5}$ increase in all the 4 cases due to duff emissions, which improves the overall model performance, although

the simulations still underestimate in the Oke11 and Oke17 cases, and slightly overestimates the $PM_{2.5}$ level in the App16 case.

The model is able to reproduce the spatial distributions of surface ozone for all fire cases (Fig. S7). The baseline (sim_nofire) runs capture the observed background daytime ozone concentrations and the concentrated $PM_{2.5}$ spots (the spots with high-level surface $PM_{2.5}$ concentrations directly due to fire smoke) in the fire and remote areas. For example, the model reproduces

the high ozone concentrations from northern Alabama and Georgia to northwestern South Carolina and North Carolina and

eastern Tennessee, as well as the coastal Louisiana and Mississippi and central Florida from Oke07 (Fig. S7f). However, the model overestimates surface ozone in the western South Carolina and North Carolina mountains (Fig. S7c and Fig. S7g). This might be caused by the uncertainty of estimating biogenic VOC emissions.

The observed ozone maximum 8-hour average (MDA8) shows an agreement with the baseline simulation for all fire cases. The observation-simulation correlation coefficients are larger than 0.5 for App16 and larger than 0.6 for the Okefenokee cases (Fig. S4). Both sim_FINN and sim_FINN+duff simulations also agree with the observations in terms of the time series and trend during the fire events (Fig. 5). The model overestimates night-time ozone by approximately 10 ppb for App16, indicating the potential bias on night-time ozone chemistry or planetary boundary layer height estimation (Li and Rappenglueck, 2018).


While the sim_FINN simulations underestimate $PM_{2.5}$ concentrations over the burning region, especially sites with the smoke impact, sim_FINN+duff simulations have better agreement with the observations, as shown in Fig. S5. The slope of the linear regression is 0.91 between sim_FINN+duff results and the observations, while the slope is only 0.15 between sim_FINN and observations. Although adding duff burning improves the regional simulation in terms of both the slope and the correlation

coefficient (from 0.29 to 0.56), the correlation coefficient is still low, indicating the potential spatial-temporal uncertainty of the fire emissions. The evaluations for the SA16, Oke11 and Oke17 fires are similar to the Oke07 results as shown in Fig. S5.

The model performance in simulating the spatial patterns of smoke is evaluated by the Moderate Resolution Imaging Spectroradiometer (MODIS) image product from the Terra satellite. Fig. S6 shows that the simulated smoke transport agrees

well with the satellite image of the smoke. The results in the Oke07, Oke11 and Oke17 fires also shows good agreement between the simulation and the satellite image. In the following sections, we will further discuss the spatial and temporal patterns of smoke ozone and $PM_{2.5}$.

### 3.2 The $PM_{2.5}$ emission and transport from duff burning

Here we show the improvement of model performance in simulating $PM_{2.5}$ by including duff burning emissions. The simulation

results on selected dates of November 15, 2016, May 10, 2007, May 8, 2011, and April 29, 2017 for the four fire cases are shown in Fig. 6 and Fig. 7, and those on other days are provided in Figs. S8 to S15.

The sim_FINN simulated smoke from App16 is transported southeastward to Georgia, South and North Carolinas on November 15, 2016 (Fig. 7a and Fig. 7e), leading to increased air pollution. However, the model underestimates the observed

surface $PM_{2.5}$ concentrations by approximately 50% in areas with peak local concentration (Fig. 6e). On November 10 and 16, 2016, the simulated plume moves in the clockwise direction, causing air pollutions in the large cities in Georgia (Fig. S9). Fig. 7a and Fig. 7e indicate that the $PM_{2.5}$ concentrations from duff burning are at the same magnitude as or even slightly higher

than those from the emissions of above-ground fuel burning. Thus, implementing duff burning doubles the fire-induced $PM_{2.5}$ concentrations during App16 over the study domain.


The total burned area of Oke07 was 5 times more than that of App16, and over the periods of the simulations, the daily average fire emission during Oke07 was 3 times more than that during App16 (Fig. S1). Correspondingly, the simulated $PM_{2.5}$ concentrations during Oke07 are greater. In addition, different from App16 that occurred in November, Oke07 occurred in May. Thus, the photochemistry of ozone and its precursors was more active. In the sim_FINN+duff runs, the simulated surface

$PM_{2.5}$ in the fire plume effectively approaches the underestimated regions showing greatest fire impact, but the enhancement is still not enough over some regions. For example, on May 10, the simulations including duff emissions are in better agreement with the observation over southwestern Florida, where the simulated concentrations are underestimated by a factor of 2-5 (Fig. 6f and Fig. 6j). Over the fire impacted region (24° N - 34° N, 76° W - 86° W) on May 10, the surface $PM_{2.5}$ increase due to duff burning is 126% more than that due to above-ground fuel burning. However, the simulation that is the closest to the

observation still underestimates the surface $PM_{2.5}$ concentrations in the fire impacted region in northern Georgia and North Carolina. The sim_FINN simulation underestimates some concentrated $PM_{2.5}$ during the fire, including southwest Florida on May 11, the Atlanta region on May 16, and western Georgia on May 26, by as much as more than 10 times sometimes (Fig. S9).

Similar to the other cases, the sim_FINN+duff simulated surface $PM_{2.5}$ concentrations from Oke11 and Oke17 are approximately doubled over the fire areas of those simulated in sim_FINN (Fig. 7c, 7d, 7g and 7h). However, the sim_FINN simulation of the fire cases does not underestimate $PM_{2.5}$ as much as Oke07. Because a large portion of the two fires was burned by the previous fires in 2007 (and 2011 for the 2017 fire), the simulated duff impacts from them are weaker. In addition, the simulated smoke from the two fires is transported to the ocean during half of the major burning periods (May 9 - 11, 2011

and May 2 - 12, 2017) (Fig. S14 and Fig. S15), which weakens the fire impact in the land areas. This inter-case comparison over the same area supports the evidence that the underestimation of $PM_{2.5}$ in the sim_FINN runs is mainly due to the missing of duff burning emissions.

The important $PM_{2.5}$ impacts of duff burning are also seen in the temporal variations of stational surface concentrations (Fig.

8). The panels 7a, d, g and h show the locations close to the fire areas. During App16, the simulated $PM_{2.5}$ concentrations increase by approximately 100% during the major burning days on November 7, 8, 13, 14 and 15, due to including duff burning (Fig. 8a). The daily variations are different between observations and simulations because the observed fire emission dataset was at daily rather than hourly intervals. The sim_FINN+duff improves the simulations of $PM_{2.5}$ surface concentrations in metro Atlanta, Georgia (Fig. 8b) and metro Charlotte, North Carolina (Fig. 8c) on major burning days.


During Oke07, including duff burning makes the simulated $PM_{2.5}$ levels 1 to 10 times closer to the observed $PM_{2.5}$ levels at many observation locations, for example, on May 9 and May 18 in Duval County, Florida (Fig. 8d), and May 8 to May 13 in Orange County, Florida (where metro Orlando is located) (Fig. 8f). The simulation shows a high bias on May 27 in Duval, potentially due to the bias on fire emissions, but on the same day, the simulation still underestimates the $PM_{2.5}$ concentrations in Atlanta, Georgia. Duff emission increases the $PM_{2.5}$ concentrations in Atlanta (Fig. 8e), but the model underestimation still exists.

Because of the weak impacts of duff burning during Oke11 and Oke17 on $PM_{2.5}$ in metro areas as shown above, we evaluated sim_FINN+duff model performance by comparing with in-situ observation at the locations close to the fire site (Fig. 8g, h). Although the sim_FINN+duff simulation overestimates $PM_{2.5}$ concentrations on May 7, 2011, May 2 - 3, 2017 and May 7, 2017, adding duff burning generally reduces the $PM_{2.5}$ underestimation in sim_FINN runs on May 9, 12, 14, 2011, April 24 - 26 and May 9 - 13, 2017. Duff burning increases local PM2.5 concentration by 50-400%, depending on the above-ground fuel burning and the duff recovery conditions.

The $PM_{2.5}$ concentrations in the sim_FINN+duff runs better fit the burning-day observations during ER08 and PB11 (Fig. 12 and Fig. 13). In June 11-12, 2011, the ER08 fire smoke transported throughout North Carolina, affected urban and rural regions more than 200 km away from the burning site. Compared to the sim_FINN runs, adding duff burning enhanced $PM_{2.5}$ by 2 to 3 times in both the Charlotte metro, North Carolina and the rural region close to burning. The $PM_{2.5}$ impact from the PB11 fire is relatively smaller in both the sim_FINN and the sim_FINN_duff runs, because the burned area is less in PB11 than in ER08, and part of the fire smoke transported to the ocean during burning. Due to duff burning, the $PM_{2.5}$ concentration increased by approximately 10% in May 11, 2011 in the Charlotte metro, and the increase is up to 100% in the rural region close to fire.

There are large mismatches at times between observations and simulations. Both biases in fire emission calculation and smoke transport simulation should be the contributors (Li et al., 2020; Garcia-Menendez et al. 2013). In addition to the uncertainties with the FINN fire emissions and duff emission calculation described above, fires have large diurnal variations, but only daily burned area data for emission calculation were available. Despite the general agreement in spatial patterns between the simulated and satellite detected smoke plume as shown above, biases in WRF simulations of atmospheric conditions, especially wind direction and speed, would lead to shifts in both space and time of the simulated plume from its actual position.

### 3.3 The different ozone and $PM_{2.5}$ impacts from duff burning

Although the above-ground fuel burning of App16 led to a 6 - 10 ppb increase of surface ozone on November 15, 2016 (Fig. 9a) over the areas affected by fire plume, adding duff burning to the model simulation does not increase the surface ozone concentration. Over the downwind region where the ozone increase is high from above-ground fuel burning, duff burning slightly offset the ozone increase by 0 - 4 ppb. A similar minor ozone impact from duff burning is also simulated for other

days (Fig. S20). The ambient VOC concentrations are lower in November than that in summer, which provides a VOC-limited scenario in the ozone photochemistry. In this scenario, when NOx concentration is high due to the above-ground fuel burning, more NOx emissions from duff burning tend to decrease ozone concentrations (Seinfeld and Pandis, 2016). The above-ground fuel burning increases ozone concentrations by 10 - 15 ppb on November 13 - 15, but the ozone concentrations in sim_FINN+duff are very similar to those in sim_FINN, indicating that the duff burning has a neglectable impact on ozone concentrations in App16 (Fig. 11a-c). The ozone simulation agrees better with observations in the urban than in the rural fire areas, and similarly to the fire area, the ozone in the urban areas is not significantly affected by duff burning.

The simulated duff burning impact on ozone is positive during Oke07, but still smaller than the $PM_{2.5}$ impact. The above-ground fuel burning and the duff burning increase the ozone concentration in the fire-impacted areas (Fig. 10b and 10f, Fig. S21). Oke07 occurred in summer and the fire site was located further south in comparison with App16, meaning higher temperature, sunlight and biogenic activities. Thus the overall VOC concentrations and the simulated ozone pollution are stronger than those in November 2016 (Seinfeld and Pandis, 2016). However, the ozone impact is significantly weaker from duff burning than from above-ground fuel burning (Fig. 10b and 10f, Fig. S21). The simulated surface ozone increase due to duff burning is only 32% of that due to above-ground fuel burning over the fire impacted region (24° N - 34° N, 76° W - 86° W) on May 10, 2007. This contribution is much smaller than that of duff burning to the $PM_{2.5}$ impact, which is 126% more than that from above-ground fuel burning. These different contributions of duff burning to $PM_{2.5}$ and ozone are due to the larger $PM_{2.5}$ emission factor but smaller $NO_x$ emission factors of duff in comparison with the above-ground fuel, as assumed in Section 2.4. The difference is also seen in the temporal variations. The above-ground burning led to ozone increases by 10 - 20 ppb in Atlanta on May 21 - 23 (Fig. 11e) and by 2 - 15 ppb in Orlando on May 8 - 12 (Fig. 11f), but duff burning led to ozone increases by 0 - 7 ppb in both Duval (Fig. 11d) and Orlando (Fig. 11f) on May 8 - 12.

The ozone increase is significant due to the above-ground fuel burning from Oke11 and Oke17 (Fig. 10c and Fig. 10d), with a level comparable to Oke07. However, the level of ozone increase due to duff burning is low (Fig. 10g and Fig. 10h). This low level is also seen in the temporal variations (Fig. 11g, Fig. 11h). The sim_FINN and sim_FINN+duff runs accurately capture several ozone peaks on May 8 and May 10, 2011, May 2 and May 8 - 13, 2017. Duff's contribution to the ozone peak is weak, similar to that of $PM_{2.5}$. The duff layer in the Okefenokee swamp in 2011 and 2017 was not well recovered from Oke07.

The ozone and $PM_{2.5}$ impacts during the simulated fire periods (App16: November 7-20, 2016; Oke07: May 6-30, 2007; Oke11: May 6-14, 2011; Oke17: April 20 - May 13 ,2017; ER08: June 8-14, 2008; PB11; May 6-14, 2011) from duff burning and the above-ground burning in the fire areas (6° x 6° in size centered at the fire site) and nearby areas are summarized in Table 3. The above-ground fuel burning significantly increases ozone over the fire area in all cases except 2017, but duff burning does not affect ozone concentrations significantly. However, duff burning has comparable $PM_{2.5}$ impacts to above-

ground fuel burning in all the fire cases. Duff burning also significantly affects urban air quality during App16 and Oke07. During 2007, when duff burning in the simulation is strong in the Okefenokee swamp, the duff impact accounts for double

that of the above-ground fuel impact. During Oke11 and Oke17, the duff impacts are weaker due to the slow recovering speed of the duff layer after the 2007 fire, but the $PM_{2.5}$ impact is still significant over the fire area.

### 3.4 Sensitivity runs

The result from Exp_2x_duff_NOx (Figure S26) shows that doubling the $NO_x$ emissions from duff does not change the result of the different $PM_{2.5}$ and ozone effects from duff burning. During the App16 case, increasing $NO_x$ further decreases the ozone concentration in the nearby urban and the closest big city. Compared to the sim_FINN runs, ozone decreases 7.5% in Charlotte, North Carolina during the App16 case, and 7.9% in the rural region of Macon, North Carolina that is close to fire. During the Oke07 case the ozone increase with more $NO_x$ from duff, the ozone in the Exp_2x_duff_NOx case is 1.3% and 4.8% more

than the sim_FINN runs over the rural region close to fire (Duval, Florida) and over the nearby big city (Orlando, Florida), still weaker than the $PM_{2.5}$ effect.

The result from Exp_FINN shows that doubling FINN emission does not affect our conclusions, as shown in Fig. S27. When the regional underestimation for $PM_{2.5}$ is 36% with no duff burning, doubling FINN emission improved the underestimation

to 20%, but still significantly underestimated the regional fire impacts. Doubling FINN emission did not fix the missing of some fire peaks on date like May 8, 11 and 14. All four fire cases shown in Fig. S27 overestimated the $PM_{2.5}$ on May 12, potentially due to the model bias on fire emission time and the smoke transport.

The result from Exp_duff (Fig. S24-25) shows that the uncertainty of duff emission does not affect our finding of the different

$PM_{2.5}$ and ozone effects by duff burning. The $PM_{2.5}$ concentrations change by -11.7% to 9.7% near the fire site for App16, and -38% to +25% for Oke07 (Fig. S24a and Fig. S24c). The ozone concentrations change within ± 2 % (Fig. S25). The $PM_{2.5}$ concentrations change by -4.6% to +2.3% in Atlanta, Georgia for App16, and -13% to +40% in Orlando, Florida for Oke07 (Fig. S24b and Fig. S24d).

**4 Conclusions and discussion**

Duff burning emissions have been calculated from the largest wildfires in this century in the moist southeastern US based on our previous field measurements at the site of the Rough Ridge Fire, one of the fires investigated in the study, and atmospheric $PM_{2.5}$ and ozone concentrations have been simulated using WRF-Chem with the duff burning emissions added to the FINN

fire emission inventory. The results indicate that contributions from duff burning to the air pollutions are comparable, and

sometimes more than the burning of above-ground fuels, which supports the previous finding from a study of the Rough Ridge Fire (Zhao et al. 2019). The WRF-Chem simulations of all the fire cases including duff burning show better agreements with the observed $PM_{2.5}$ surface concentration than the baseline simulations which include only fire emissions from above-ground fuel burning. Thus, regional air quality modeling in the southeastern US can be substantially improved by adding duff burning emissions in the existing fire emission datasets. It is further concluded that the impacts of duff burning on $PM_{2.5}$ are much more

remarkable than those on ozone. The simulation results indicate that the above-ground fuel burning increases regional ozone surface concentrations, but the ozone changes due to duff burning are statistically insignificant.

The importance of duff burning contribution to $PM_{2.5}$ concentrations suggests an effective approach to improve regional air quality simulations in the other global regions with deep and peat duff accumulations. As described before, current major fire

emission inventories, such as GFED and FINN (Wiedinmyer et al., 2011; Giglio et al., 2013; Van Der Werf et al., 2017), do not include enough duff and peat emissions. FINN v1.5 applied the emission factors mainly based on Andreae and Merlet (2001) and Akagi et al. (2011) but the emission factors of duff and peat burning are not included. On the other hand, Tansey et al. (2008) investigated the uncertainties of burned area and satellite fire hotspots over the tropical peatlands, indicating that duff burning in the emission inventories based on satellite data is highly uncertain. This potentially leads to significant

underestimation over fire events with duff and peat burning, which further affects the evaluation of the regional air quality and human health impacts (Reid et al., 2016).

One major uncertainty in this study is the amount of duff burned in the Okefenokee fires. Although reports show that during the 2007 Okefenokee extreme drought and fire, 2 feet (61 cm) of duff thicknesses reduction was observed in some intensively

burned areas, which potentially due to the combination of burning and deflation of domed duff and peat surfaces, an average duff consumption for simulation is not provided from measurements (Johnson and Schmerfeld, 2016). In the sensitivity runs described in Section 3.4, we indicate that this bias has a minimal effect on the major findings of the large $PM_{2.5}$ impact from duff burning and the different impacts between $PM_{2.5}$ and ozone. Another uncertainty is the values used in the duff emission calculation, for example, the bulk density of duff mass and the emission factors of different species from duff burning. We

used $50 \pm 16$ g/kg $PM_{2.5}$ emission factor for duff burns in this study based on fires in North Carolina (Geron and Hays, 2013; Urbanski, 2014), which is closest to the fire sites. It is comparable to some other peat fire measurements in the southeastern US such as $44 \pm 9$ g/kg from Benner (1977) and $30 \pm 20$ g/kg from McMahon et al. (1980). However, the spatial variability of duff bulk density is large in the southern US, ranging from 39.4 to 103.7 kg $m^{-2}$ $m^{-1}$ (Ottmar and Andreu, 2007).

Many evaluation studies have indicated that WRF-Chem is able to provide ground ozone simulations within reasonable biases, less than 20% for Europe (Mar et al., 2016) and 15-30% for the western, northeastern and midwestern US (Astitha et al., 2017). However, ozone simulation within plume is much more complexed, depending on many factors such as emissions of ozone

precursors, photochemical processes, radiation change and temperature changes due to smoke, and lifetime of smoke, which make simulate ozone from wildfires challenging (Jaffe and Wigder, 2012). Our simulations did not consider the impacts of smoke on radiation, possibly leading to overestimating ozone production in plume (Selimovic et al., 2020). We conducted a test simulation for the Ofe07 case by including the impacts to evaluate the related uncertainty in ozone simulations. The result shows that missing the aerosol radiation impacts leads to approximately 15% of ozone overestimation in the fresh plume, and 10% of ozone increase in the aged plume. Further evaluation of ozone simulation in fire plume is needed. The recent implements of many field campaigns, especially Western wildfire Experiment for Cloud chemistry, Aerosol absorption and Nitrogen (WE-CAN) (https://www2.acom.ucar.edu/campaigns/we-can), are expected to help fill the evaluation and simulation gaps.

The findings from this study on the air quality impacts of wildfires in the southeastern US are valuable for future studies and can serve as guidance for other global regions with duff and peat burning such as northeastern China (Jiang et al., 2008) and the Great British Isles (Davies et al., 2013). In the southeastern US, the general high humidity provides good conditions for the duff layer to accumulate, which serves as a large potential fuel source during wildfires under droughts. The peatland in boreal forests (e.g. the boreal forest in Canada and Northern Eurasia) and tropical forests (e.g. the peatland in Indonesia) are also vulnerable to fire. Although the duff and peat layer and the burning types vary with ecosystems, the carbon loss from duff and peat fire and the different emission factors between the below-ground fuel and above-ground fuel are common issues (Page et al., 2002; Turetsky et al., 2015), which need to be addressed as what was conducted for the southeastern US in this study. The $PM_{2.5}$ emission factor used in this study is higher than the measurements in the other regions, such as 20.6 g/kg estimated in the US prescribed burning (Yokelson et al., 2013), 8-58 g/kg measured in fires in Southeast Asia (Roulston et al., 2018), and 18.9 g/kg from global estimation summarized by Andreae (2019). This difference suggests that the impacts of duff burning during flaming phase on $PM_{2.5}$ may be more remarkable in the southeastern US than many other world regions.

Duff consumption in different fire cases is highly variable, making it difficult to conduct practical operational prediction of duff consumption and the air quality impacts. A number of efforts could be made towards a solution. One is to map spatial distributions of duff. Fuel data such as the Fuel Characteristics Classification System (Prichard et al., 2019) could be expanded to include more complete duff information. The data need to be dynamical to reflect not only duff accumulation over time but also disturbance due to wildland fires. Another effort is to conduct more field measurements of duff consumption by both wildfires and prescribed fires, such as those by Zhao et al. (2019) and the Fire and Smoke Model Evaluation Experiment (FASMEE) (Prichard et al., 2019). The measurements are essential for developing tools for duff consumption and air quality impact modeling (Liu et al., 2019). Duff burning by flaming fires occurs mainly under persistent drought conditions. Thus, duff fuel moisture is a critically important parameter to predict if and how much duff will be consumed by a wildfire. There are fire danger rating systems such as the Canadian Forest Fire Weather Index (FWI) System (FWI) (Stocks et al., 1989) and FARSITE (Finney, 1998) that estimate duff fuel moisture. They are empirically based rather than physics based dynamical

tools. Improvements to these tools and development of dynamical tools, including those that relate duff fuel moisture with drought indices such as the Keetch-Byram Drought Index (KBDI) (Keetch and Byram, 1968), are needed. In this study, we focused more on duff flaming than smouldering because of the relatively weak ability to transport of smouldering and the

limitation of WRF-Chem to well processes smouldering. We are planning to dig into the duff smouldering phase more in a separate study using a specific smoke model such as the PB-P model (Liu et al., 2018)

Under climate change due to the increasing atmospheric greenhouse gases, duff burning becomes more important for PM simulation and the air quality impacts. Duff burning is likely to become more active under the changing climate. The increasing

frequency of extreme droughts has been observed in the US (Mazdiyasni and AghaKouchak, 2015; Clark et al., 2016) and around the world, and projected for the future climate scenario (Masih et al., 2014; Longo et al., 2018; Grillakis, 2019). Therefore, fire events ignited on a generally wet land suffered by extreme drought are likely to happen more often in the future, and the duff and peat land that does not burn currently (e.g. the Amazon rainforests and Africa rainforests (Bonal et al., 2016)) may become burnable under future extreme drought. The importance of duff burning is further strengthened with climate-

ecosystem interactions. With the increasing mean temperature and $CO_2$ concentrations, the duff layer accumulation is potentially benefiting from the acceleration of vegetation growth (Qian et al., 2010; Huang et al., 2018; Lawal et al., 2019; Bai et al., 2020) and soil organic carbon decomposition (Fierer et al., 2006; Karhu et al., 2014). Besides, tropical peatland fires are sensitive to ENSO-induced climate variability, indicating that it is necessary to evaluate the fire-climate interactions in order to better understand the duff and peat burning (Field et al., 2009; Tosca et al., 2011).


*Data availability.* The EPA AQS measurement data of ozone and $PM_{2.5}$ are available at the EPA website (https://www.epa.gov/outdoor-air-quality-data, last access: October 22, 2020). The FINNv1.5 fire emission inventory is available at the NCAR ACOM website (https://bai.acom.ucar.edu/Data/fire/, last access: January 16, 2021). The MOZART simulation results used for simulation initial and boundary conditions are available at https://www.acom.ucar.edu/wrf-

chem/mozart.shtml, last access: January 16, 2021. The WRF-Chem model results are available from the corresponding author upon request.

*Competing interests.* The authors declare that they have no conflict of interest.

*Author contribution.* Yongqiang Liu provided the original idea. Aoxing Zhang and Yongqiang Liu designed the model experiments. Aoxing Zhang carried out the model experiments, analyzed the observation and simulation data and prepared the manuscript. Yongqiang Liu, Scott Goodrick and Marcus D Williams contributed to the methodology and manuscript improvement.

*Acknowledgments.* This study was supported by an agreement between the USDA Forest Service and the Oak Ridge Institute for Science and Education (ORISE).

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

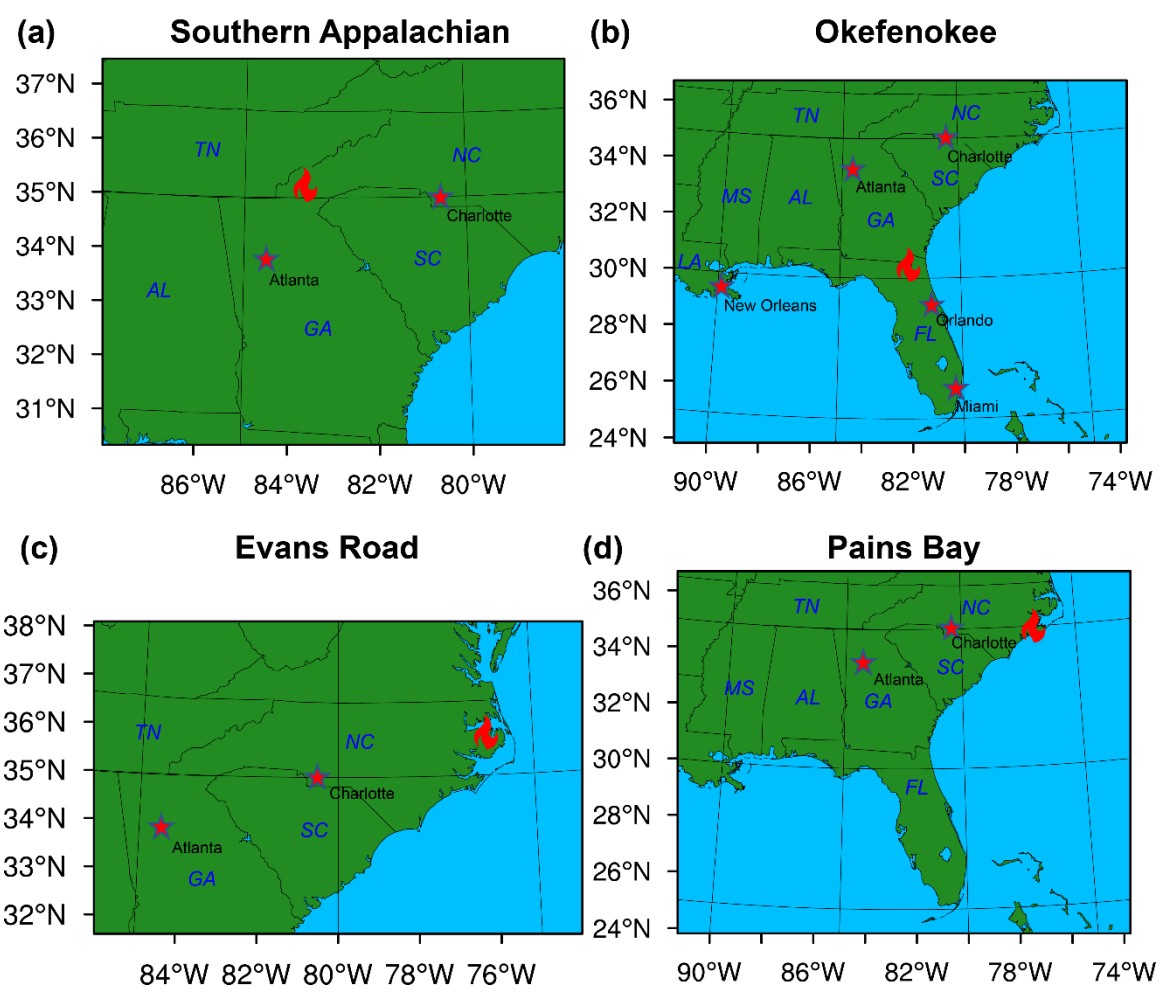

**Figure 1. The WRF-Chem domain used in (a) App16, (b) the Okefenokee cases, (c) ER08 and (d) PB11. The fire sites and nearby major cities are marked. The abbreviations of the US state names are Florida (FL), Alabama (AL), Georgia (GA), South Carolina (SC), North Carolina (NC), Tennessee (TN), Mississippi (MS) and Louisiana (LA).**

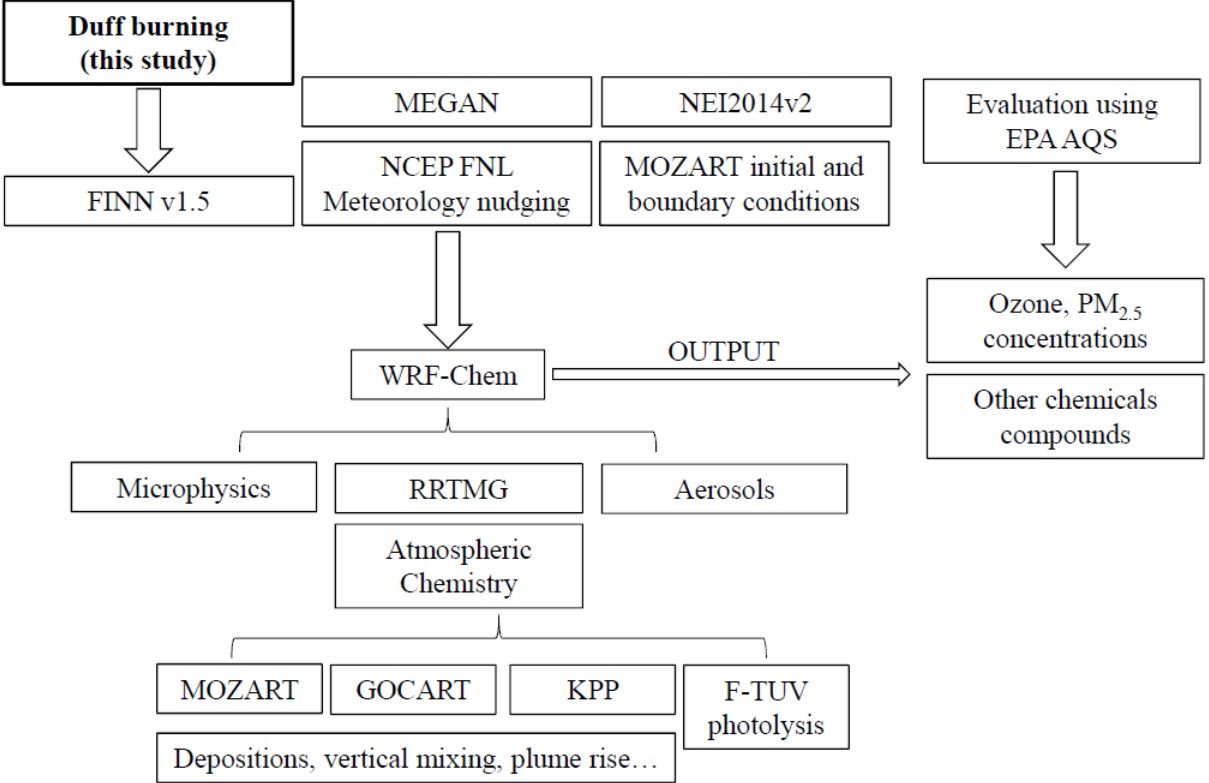

**Figure 2. Description of the model components, input data, and implementation procedures.**

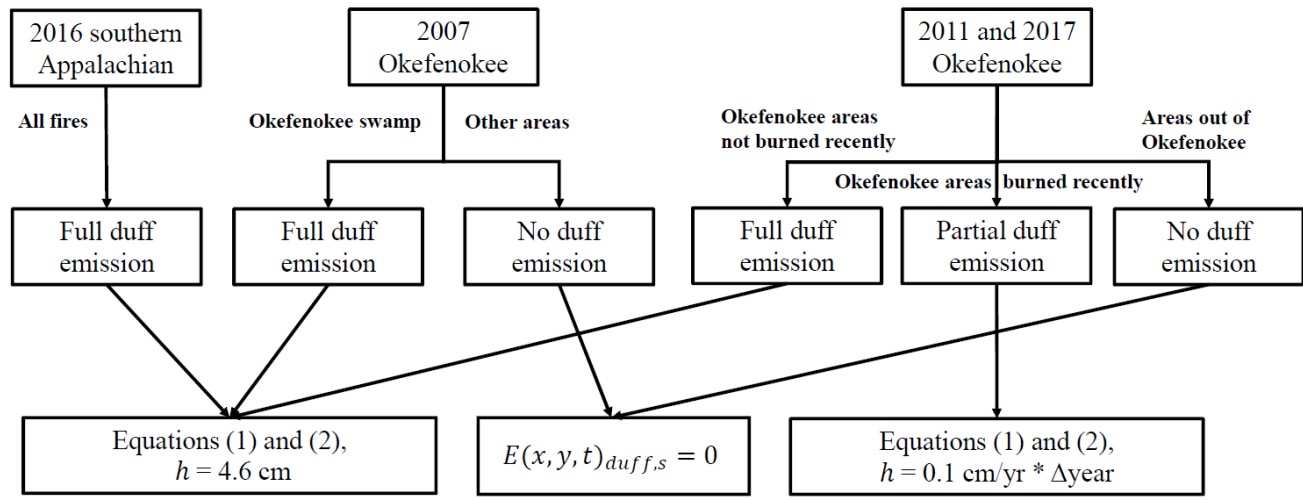

**Figure 3. Description of the duff emission estimation.**

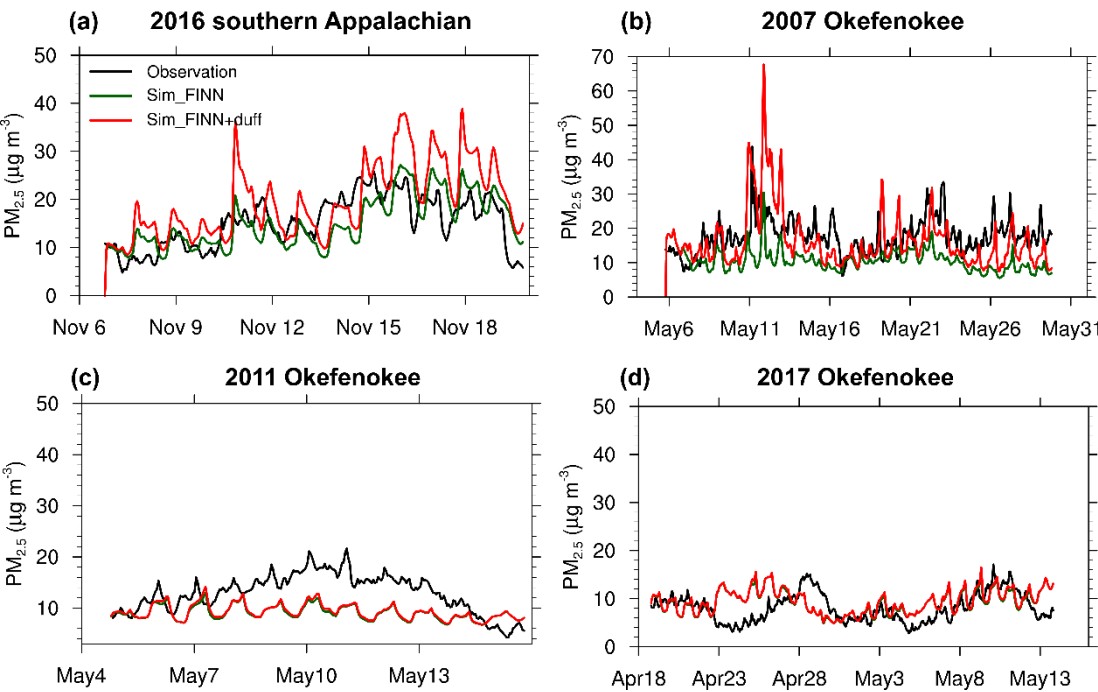

**Figure 4. The time series of hourly surface PM₂.₅ concentrations. Black: Measurements averaged over observation sites within the simulation domain. Green and red: Simulations of Sim_FINN and Sim_FINN+duff, respectively, averaged over the observation**
**sites.**

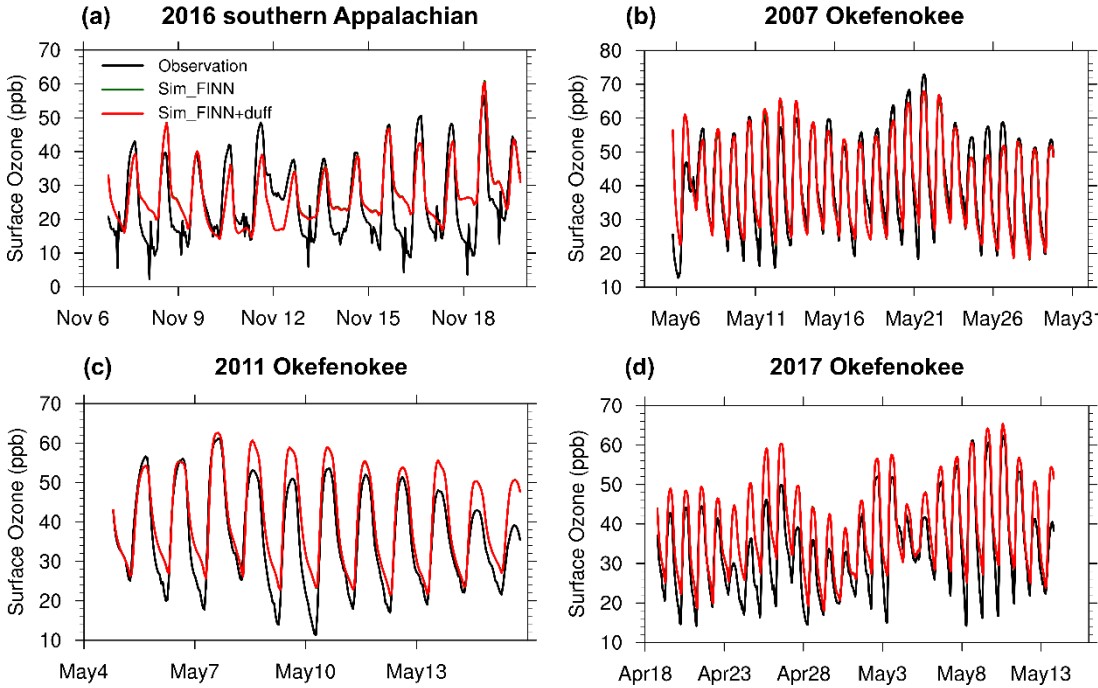

**Figure 5. The time series of hourly surface ozone concentrations. Black: Measurements averaged over observation sites within the simulation domain. Green and red: Simulations of Sim_FINN and Sim_FINN+duff, respectively, averaged over the observation sites.**

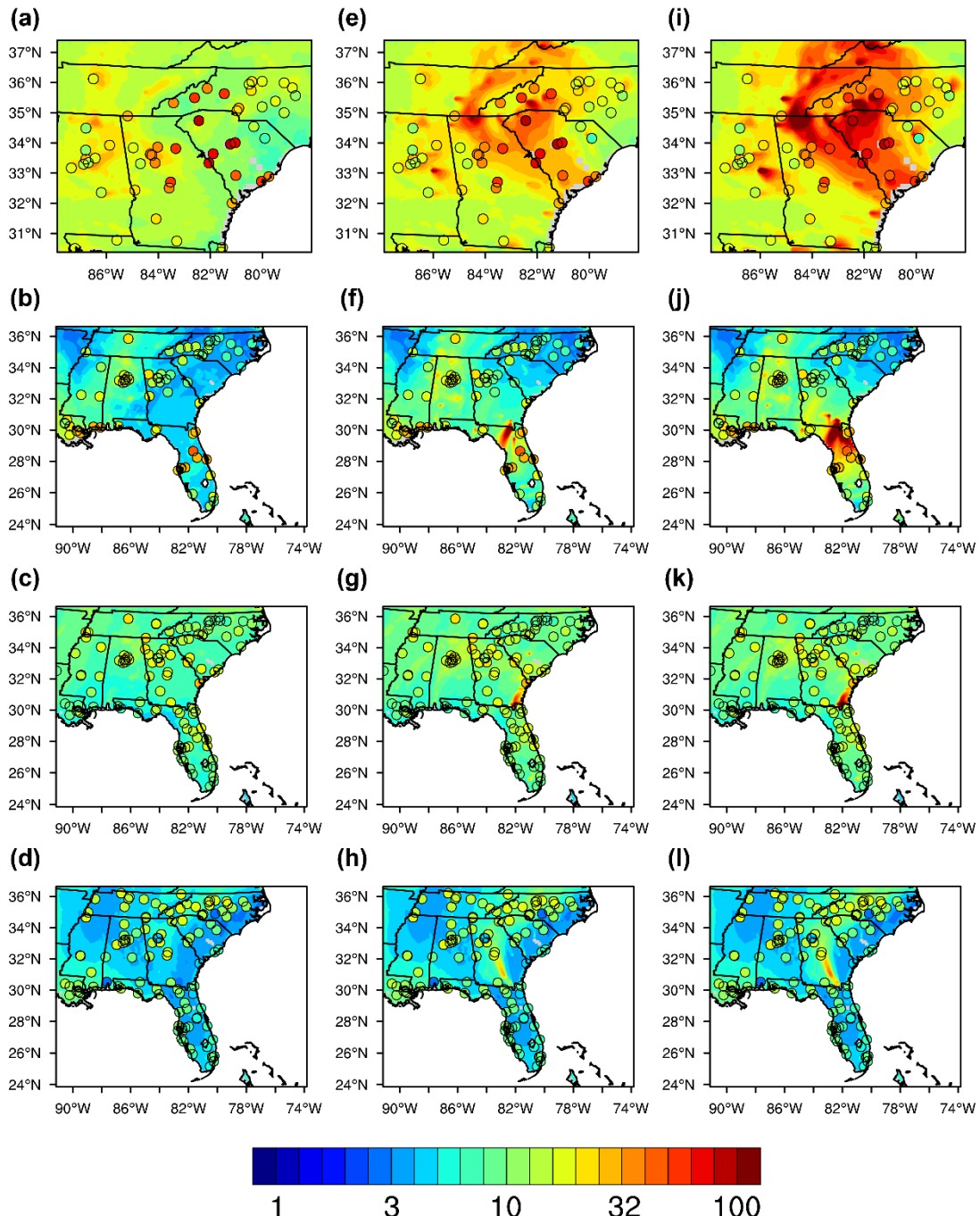


**Figure 6. The mean surface concentration of simulated and observed PM2.5 (μg m-3) on representative days. (a) App16 (November 15, 2016), (b) Oke07 (May 10, 2007), (c) Oke11 (May 8, 2011), and (d) Oke17 (April 29, 2017) for sim_nofire. (e) - (h) are the corresponding fire cases for sim_FINN, and (i) - (l) are the corresponding fire cases for sim_FINN+duff. The color scatters represent the observed daily mean PM2.5 concentrations.**

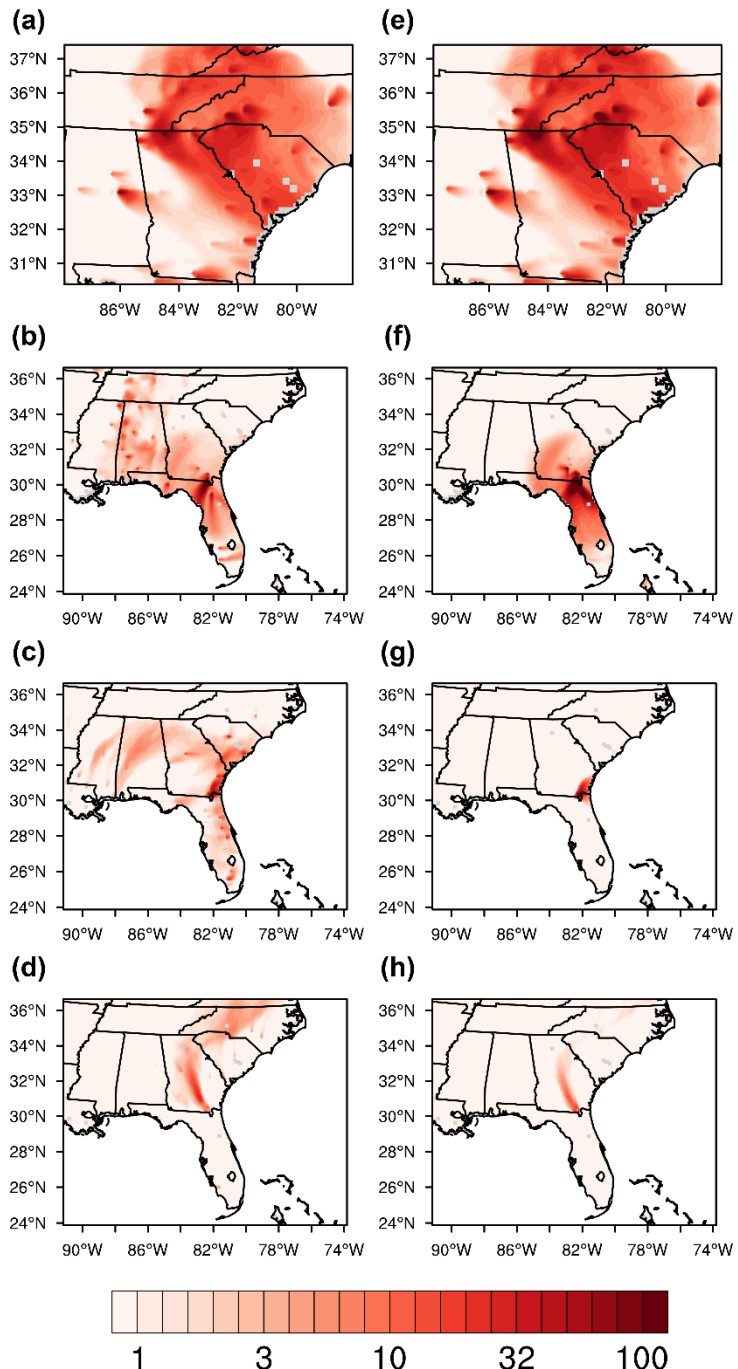

Figure 7. The surface PM2.5 concentration change (μg m-3) on representative days. (a)   The change due to above-ground fuel burning during App16 (November 15, 2016), (b) Oke07 (May 10, 2007), (c) Oke11 (May 8, 2011), and (d) Oke17 (April 29, 2017). (e) - (h) are the corresponding changes due to duff burning.

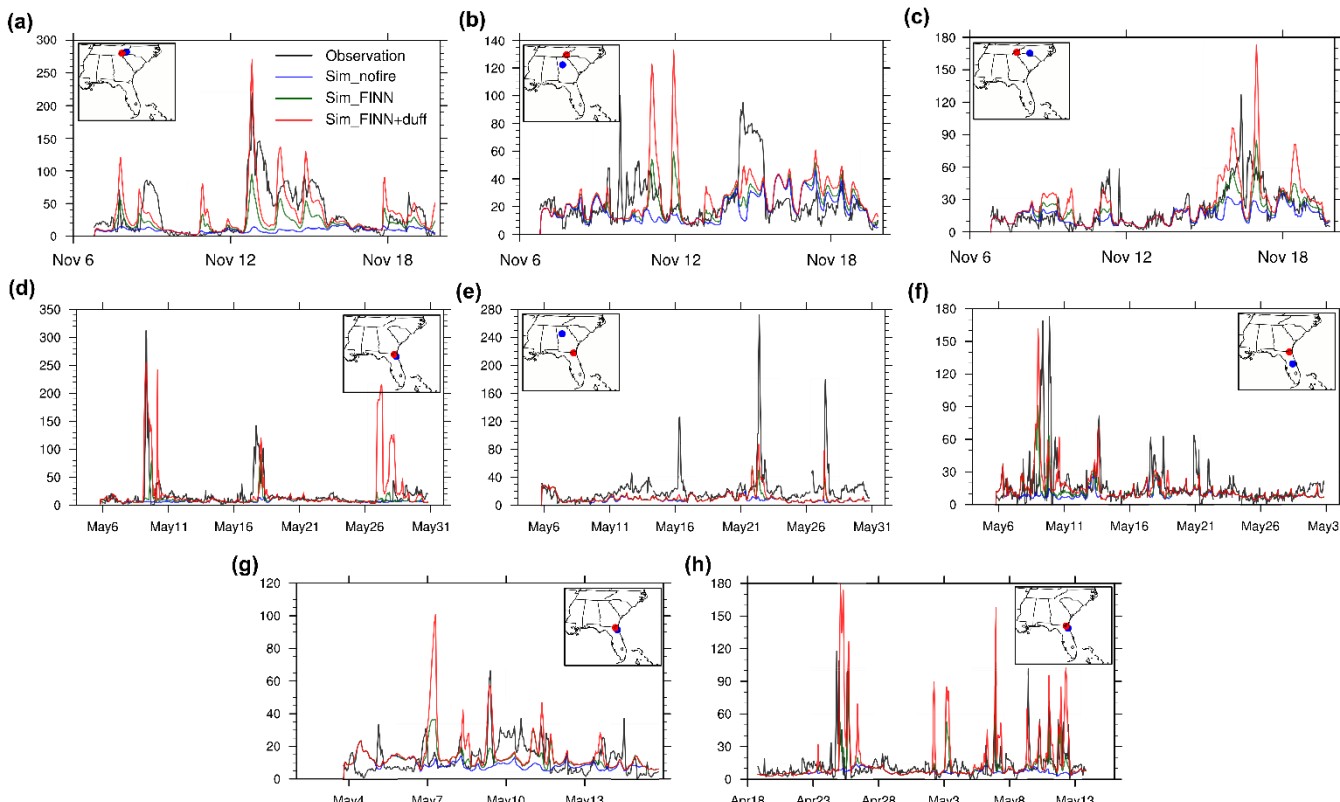

**Figure 8. Comparisons of in-situ hourly surface PM2.5 concentrations (µg m-3) among the observation (black), sim_nofire (blue), sim_FINN (green) and sim_FINN+duff (red) simulations. (a - c) App16, (d - f) Oke07, (g) Oke11 and (h) the 2017 Okefenekee Fire. The fire location (red) and site location (blue) are shown in the map attached to each panel. The observation sites are located in (a) Macon county, NC, (b) Fulton county, Georgia, (c) Mecklenburg county, NC, (d) Duval county, Florida, (e) Fulton county, Georgia, (f) Orange county, Florida, (g) - (h) Duval county, Florida.**


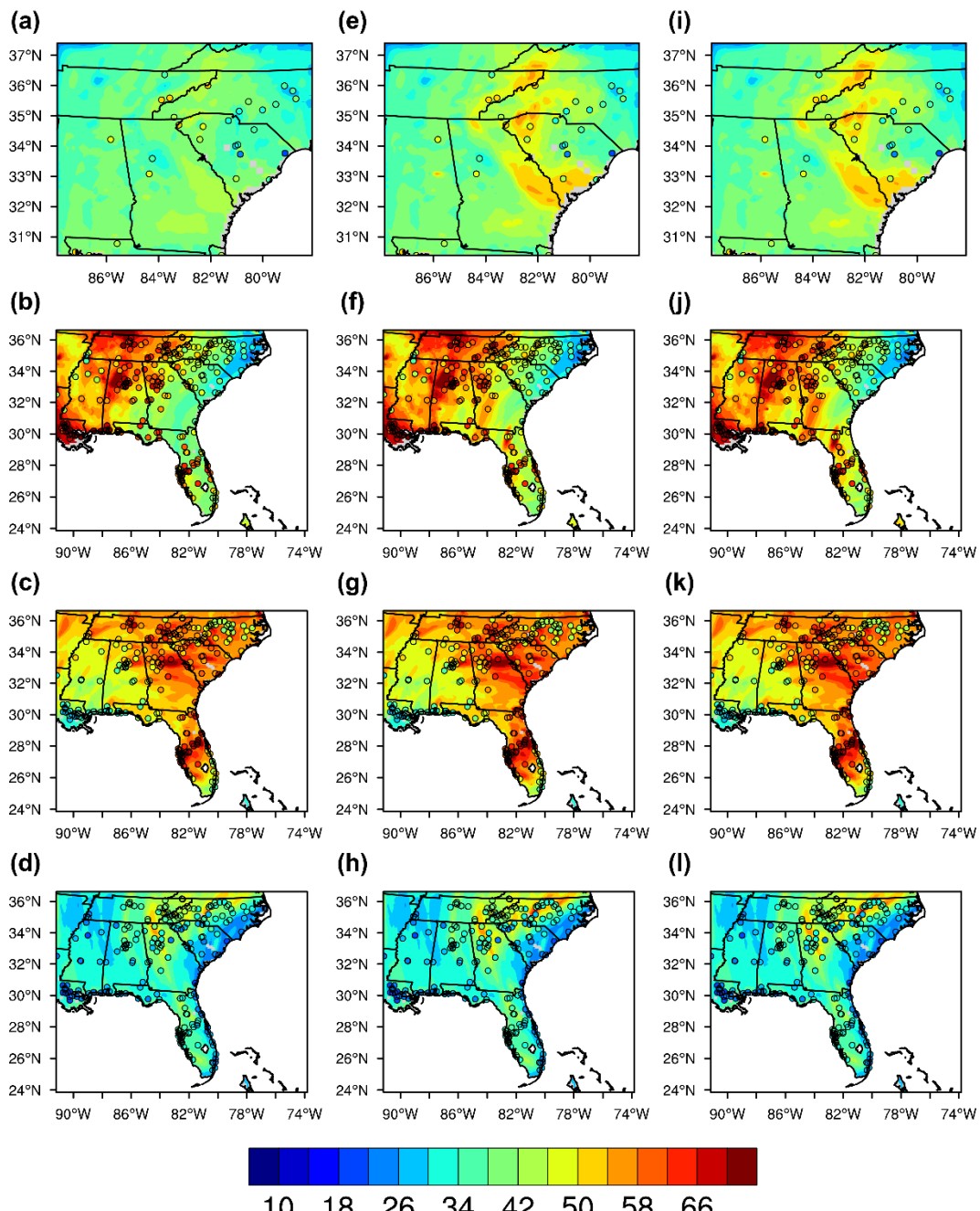

**Figure 9. The day-time mean (from local time 10 am to 6 pm) surface ozone concentration of simulated and observed ozone (ppb) on representative days. (a) App16 (November 15, 2016), (b) Oke07 (May 10, 2007), (c) Oke11 (May 8, 2011), and (d) Oke17 (April 29, 2017) for sim_nofire. (e) - (h) are the corresponding fire cases for sim_FINN, and (i) - (l) are the corresponding fire cases for sim_FINN+duff. The color scatters represent the observed day-time mean surface ozone concentrations.**

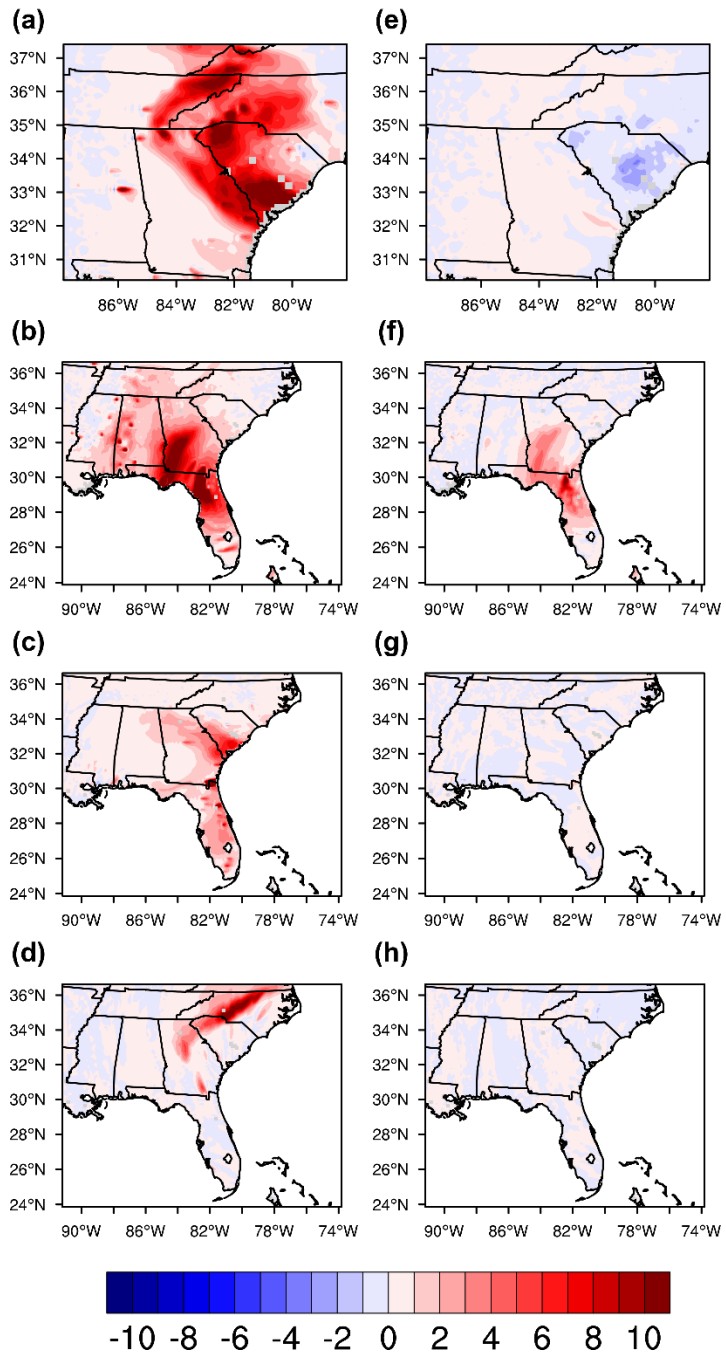

**Figure 10. The day-time mean (from local time 10 am to 6 pm) surface ozone concentration change due to above-ground fuel burning (ppb) on representative days. (a) 2016 southern Appalachian case (November 15, 2016), (b) Oke07 (May 10, 2007), (c) Oke11 (May 8, 2011), and (d) Oke17 (April 29, 2017). (e) - (h) are the corresponding changes due to duff burning.**


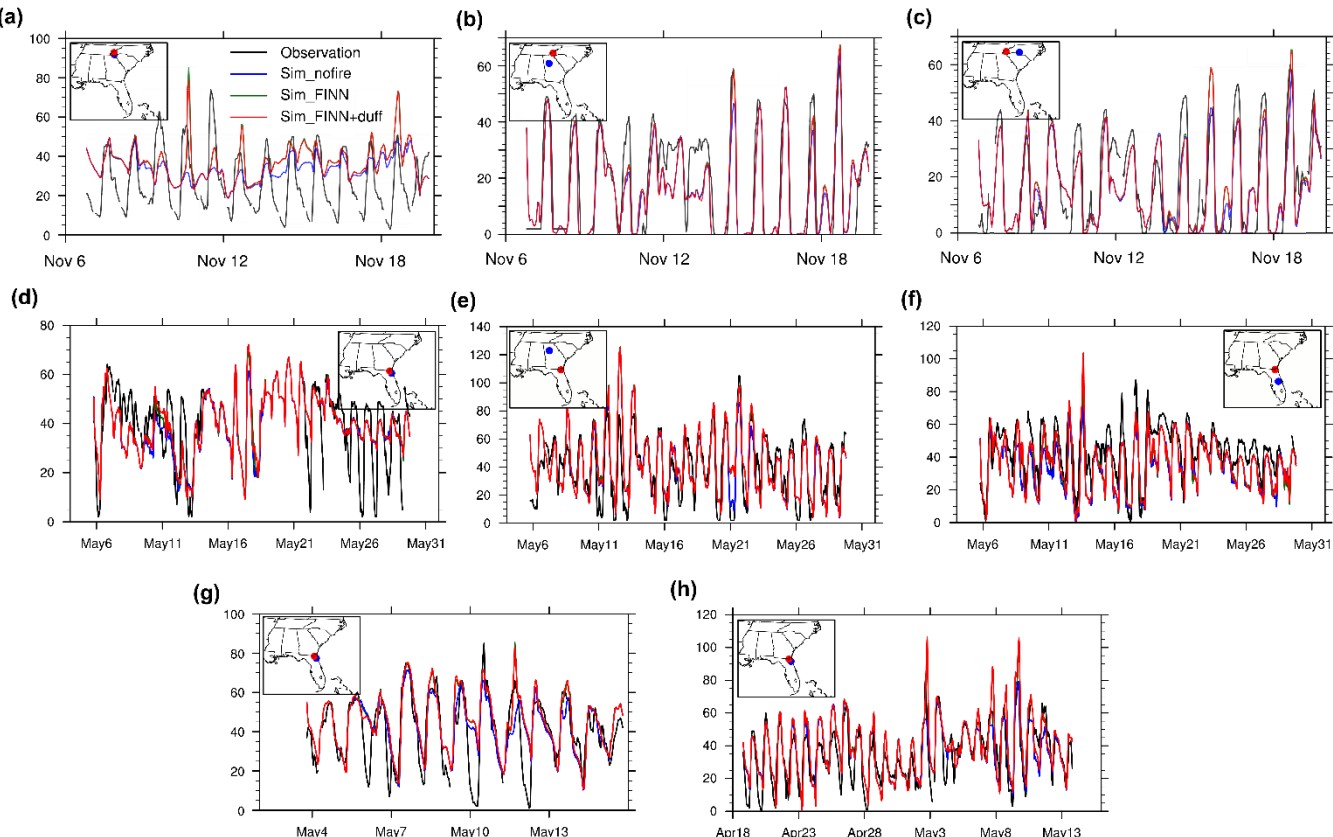

**Figure 11. Comparisons of in-situ hourly surface ozone concentrations (ppb) among the observation (black), sim_nofire (blue),**
**sim_FINN (green) and sim_FINN+duff (red) simulations. (a - c) App16, (d - f) Oke07, (g) Oke11 and (h) Oke17. The fire location**
**(red) and site location (blue) are shown in the map attached to each panel. The studied sites are in (a) Macon county, North Carolina,**
**(b) Fulton county, Georgia, (c) Mecklenburg county, NC, (d) Duval county, Florida, (e) Fulton county, Georgia, (f) Orange county,**
**Florida, (g) - (h) Duval county, Florida.**

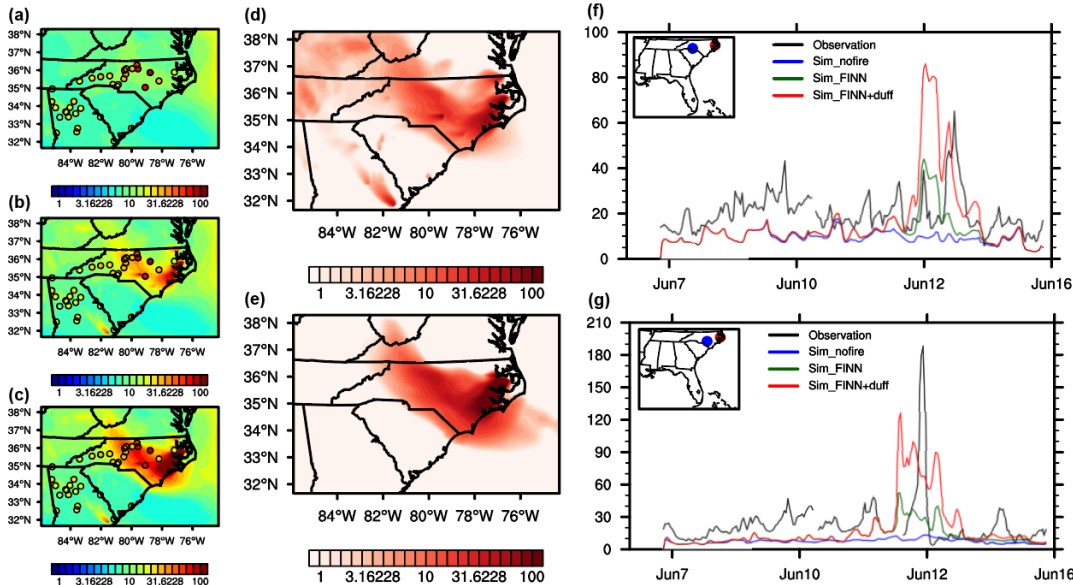

Figure 12. The PM2.5 distribution and time series during the 2008 Evans Road Fire. (a-c) PM2.5 daily mean surface concentration (µg m-3) on June 12, 2008 simulated in (a) sim_nofire, (b) sim_FINN and (c) sim_FINN+duff runs. (d-e) The PM2.5 daily surface concentration differences (µg m-3) between (d) sim_FINN and sim_nofire and between (e) sim_FINN+duff and sim_FINN on June 12, 2008. (f-g) The comparison of the time series of hourly surface PM2.5 concentrations (µg m-3) between the observation and simulations from June 7 to June 14, 2008 in (f) Mecklenburg county, NC and (g) Cumberland county, NC.

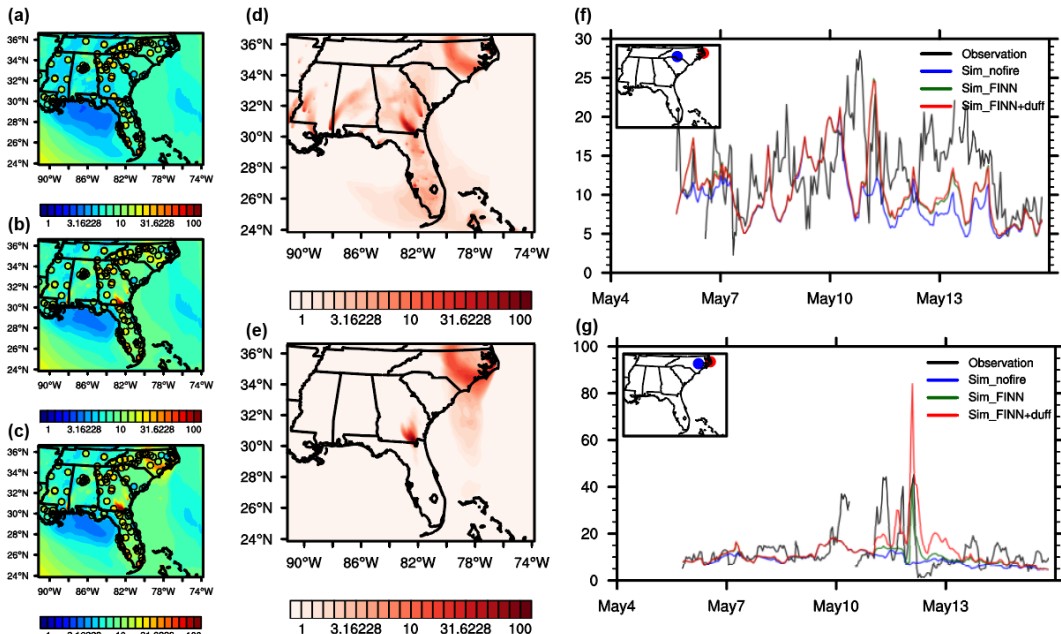

Figure 13. The PM2.5 distribution and time series during the 2011 Pains Bay Fire. (a-c) PM2.5 daily mean surface concentration (µg m-3) on May 12, 2011 simulated in (a) sim_nofire, (b) sim_FINN and (c) sim_FINN+duff runs. (d-e) The PM2.5 daily surface

concentration differences (µg m-3) between (d) sim_FINN and sim_nofire and between (e) sim_FINN+duff and sim_FINN on May 12, 2008. (f-g) The comparison of the time series of hourly surface PM2.5 concentrations (µg m-3) between the observation and simulations from May 6 to May 15, 2011 in (f) Mecklenburg county, NC and (g) Wayne county, NC.

Table 1. The simulation period and fire emission inventories applied in different WRF-Chem simulations and experiments.

| Simulation and experiment | Simulation Period | Fire emission | |
|---|---|---|---|
| | | FINN fire emission | Duff emission |
| Sim_nofire | App 16: 11/7-11/22, 2016, 91,191 acres Oke07: 5/6-5/30, 2007, >500,000 acres Oke11: 5/4-5/15, 2011, 147,065 acres Oke17: 4/19–5/13, 2017, 166,737 acres ER08: 6/7-6/15, 2008, 41,560 acres PB11: 5/4-5/15, 2011, 29,400 acres | No | No |
| Sim_FINN | | 1x FINN emission | No |
| Sim_FINN+Duff | | 1x FINN emission | 1x duff emission |
| Exp_FINN | Oke07: May 6-16, 2007, >500,000 acres | 2x FINN emission | 1x duff emission |
| Exp_duff | App16: Nov 7-14, 2016, Oke07: May 6-16, 2007 | 1x FINN emission | 0.7x duff emission |
| | | | 1.3x duff emission |
| 2x duff NOx | App16: Nov 7-14, 2016, Oke07: May 6-16, 2007 | 1x FINN emission | 2x duff emission for NOx 1x duff emission for the other species |

Table 2. Comparison of duff and temperate mixed forest emission factors (g/kg) used in this study.

| Species | Peat and duff | FINN temperate mixed forest |
|---|---|---|
| CO | 271±51[a] | 102[e] |
| NO | 0.559[b] | 0.34[e] |
| NO$_2$ | 0.176[b] | 2.7[e] |
| SO$_2$ | 1.76[b] | 1[f] |
| NH$_3$ | 2.67[b] | 1.5[e] |
| PM$_{2.5}$ | 50±16[c] | 13[f] |
| OC | 37.5[d] | 9.2[f] |
| BC | 0.375[d] | 0.56[f] |

[a] Urbanski 2014, averaged based on Geron and Hays (2013) and Hao et al. (2007).

[b] Yokelson et al. (2013)

[c] Urbanski 2014, an average of Geron and Hays (2013)

[d] An estimated 100:1 ratio of OC/BC emission factors based on Jen et al. (2019), after applying the PM2.5/carbonaceous aerosol emission ratio from the FINN emission factors.

[e] Akagi et al. (2011)

[f] Andreae (2008) in extratropical Forest

**Table 3. Summary of the increased ratio of PM2.5 and ozone due to duff burning and above-ground fuel burning. The bold numbers represent that the increase or decrease ratio passes the Student's t-test with p = 0.05.**

| Fire Case | Location | fire-nofire PM$_{2.5}$ | duff-noduff PM$_{2.5}$ | fire-nofire ozone | duff-noduff ozone |
|---|---|---|---|---|---|
| Oke07 | Fire region | **63.40%** | **131.90%** | **3.30%** | 0.90% |
| Oke07 | Atlanta | **13.20%** | **6.00%** | 2.10% | -0.10% |
| Oke07 | Charlotte | 7.20% | 2.60% | 1.40% | -0.10% |
| Oke07 | Orlando | **28.30%** | **17.70%** | 9.10% | 2.90% |
| Oke07 | Miami | **27.20%** | **24.80%** | **7.00%** | 1.90% |
| Oke07 | New Orleans | **9.80%** | **8.50%** | 4.10% | 2.20% |
| App16 | Fire region | **80.60%** | **61.30%** | **5.20%** | -0.20% |
| App16 | Atlanta | **28.10%** | **21.30%** | 10.70% | 2.50% |
| App16 | Charlotte | **41.20%** | **29.70%** | 22.50% | 4.90% |
| Oke11 | Fire region | **41.70%** | **13.00%** | **4.80%** | 0.20% |
| Oke17 | Fire region | **29.70%** | **10.90%** | 2.70% | 0.00% |
| ER08 | Fire region | **60.02%** | **137.30%** | 3.80% | 0.89% |
| PB11 | Fire region | **14.1%** | **32.7%** | 12.0% | 0.22% |

[*]The "fire region" is the squared 6° x 6° area with the fire site in the center.

1160