# Peer review of "Duff burning from wildfires in a moist region: different impacts on PM2.5 and Ozone"

_Atmospheric Chemistry and Physics, 2021_

## Author Response (AR1)

**Responses to Comments from Referee #1**

The purpose of this study is to assess the importance of including duff in simulation of wildfire impacts on air quality. The authors conduct WRF-Chem simulations for four large wildfire events in the southeastern US using two fire emission scenarios – duff and no duff, as well as a control simulation with no fire emissions. The main findings of the study are:

1) relative to the no duff fire scenario (surface and understory fuels only), the increase in wildfire emissions through the inclusion of duff burning resulted in large increases in simulated surface PM2.5 concentrations near the fire locations (< 300km) and at remote urbans areas

2) while the no duff fire scenario increased regional O3 levels, the impact of additional emissions from duff burning were negligible for O3

3) relative to the control scenario (no fire) both the no-duff and duff fire simulations generally increased agreement between the WRF-Chem simulated PM2.5 and observations at surface air quality monitoring sites in fire impacted areas

4) relative to the no duff fire scenario, inclusion of duff emissions generally improved agreement of the WRF-Chem PM2.5 and observations at surface air quality monitoring sites in fire impacted areas

The authors conclude that modeling of regional air quality in the southeastern US can be improved by adding duff burning emissions to existing fire emission datasets and that emissions from duff burning have much garter impact on PM2.5 than O3.

The topic addressed in this study, contributions of duff and peat burning emissions to regional air quality, is certainly an important and of interest to air quality modelers and atmospheric scientists and is relevant to biomass burning in many regions. However, in the manuscript requires major revisions before it is suitable for publication. The paper is missing key methodological details and a couple important choices in the study design are not well justified. Additionally, the presentation and discussion of results and lacks definition and focus, making it difficult to evaluate the authors' conclusions and the overall broader relevance of the study.

Here I provide my most important concerns regarding the paper, followed by less crucial, specific comments.

**Response:** Thank you for reviewing our manuscript and providing critical and helpful comments. We revised the manuscript based on the comments to provide missed key methodological details and focus more on presentation and discussions.

**1. Estimation of duff consumption by flaming combustion**
In Zhao et al. (2019), post-fire field measurements at 4 sites (2 pair: 2 burned, 2 unburned) at the location of the ~11,700 ha Rough Ride Fire, indicated duff depth consumption of 4.5 cm and 4.9

cm. Based on undocumented and unreferenced information, "In fact, whereas a duff layer is typically consumed during the smoldering phase of combustion, the monitoring and images taken during the RRF indicated that a large portion of the duff layer burned during the flaming phase of combustion.", Zhao et al. (2019) assume that nearly all of the duff consumption occurred during flaming combustion in one day. In the current study, the authors use the 4.5 cm duff depth consumed by flaming consumption claimed by Zhao et al. (2019) and apply it to the four fire cases. This choice does not seem justified based on the less than robust information presented in Zhao et al. (2019).

**Response:** The photos below show night fire lines during the 2016 Rough Ridge Fire. Flaming is clearly seen. The duff layer was burned during flaming phase due to a half-year long severe drought prior to the fire. One coauthor of Zhao et al. (2019), Jeffrey Schardt served as a fire manager from the Chattahoochee-Oconee National Forests, USDA Forest Service monitored the entire development of the fire. We agree that there are no robust duff burning measurements in the Southeast US although a number of large fires have occurred in the duff-rich areas. Due to this limitation of direct measurements from different fire events, we had to apply the 4.5 cm duff burning depth to other fire cases. In Section 3.4, we discussed the uncertainty and conducted sensitivity runs to evaluate the corresponding variations of PM2.5 and ozone effects due to the duff emission uncertainty. On the other hand, this uncertainty was actually one of implications of the findings from this study for future fire emission and smoke air quality. The finding indicated the importance to measure and simulate duff burning for assessing the contributions of smoke to global air pollutions.

[Figure]

**2. Duff PM2.5 emission factors**

The study is simulating the impacts of flaming duff consumption on air quality, but they use PM2.5 EF factors for smoldering duff (Urbanski 2014; Geron & Hays, 2013). The high PM2.5 and VOC emission factors for duff burning result in large part because duff burns primarily by smoldering combustion. The authors should have used a reduced PM2.5 EF to represent flaming combustion.

**Response:** Thanks for this point. In Geron and Hays (2013), the peat and duff emission factors were

measured simultaneously in lab experiments under simulated condition for peat and duff smoldering and flaming condition for other fuels (Section 3.2 in Geron and Hays 2013). There is limited information about the emission factor of duff flaming and its difference to duff smoldering, but individual and synthesis studies showed that the combustion efficiency of duff flaming is higher than smoldering (Lin et al. 2019), and that the flaming emission factor of $PM_{2.5}$ for some above-ground fuel is 30% lower than smoldering (Fig. 3 in Prichard et al. 2020). To further address the reviewer's concern, we conducted sensitivity runs tuning the duff $PM_{2.5}$ emissions by ± 30%, and discussed the potential bias and uncertainty due to duff emissions. The results of the sensitivity runs are shown in Figures S24 and S25. Despite the uncertainty from duff emission factors and fuel loading, tuning the duff $PM_{2.5}$ emissions by ± 30% did not affect the major conclusions.

In the revision, we included the uncertainty due to the different emission factors between smoldering and flaming in Line 360-362: "There were not enough duff measurements for the fire cases we investigated, and the duff emission factors between smoldering and flaming were also not well investigated.". Quantitative analysis in the Section 3.4 is also updated.

**3. Temporal emissions profile**
The authors do not describe how the daily fire emissions were converted into hourly emissions for the WRF-Chem simulation. The appear seems to imply they were not:
L408-409: "The daily variations are different between observations and simulations because the observed fire emission dataset was at daily rather than hourly intervals."

**Response:** The WRF-Chem simulations did not consider the diurnal cycle of fire emissions. FINN inventory analyzed the daily MODIS Thermal Anomalies Product and assumed the fire lasts for 1 day without diurnal cycle. Therefore, the fire emission applied in the WRF-Chem simulations for each hour is the hourly emission converted from the daily fire cases from FINN. We added the description into the revised manuscript (Line 265-267): "No a-priori diurnal cycle of the fire emission was applied in the WRF-Chem model, and the hourly fire emission applied in the WRF-Chem simulations was the hourly emission converted from the daily fire cases from FINN assuming each observed fire hotspots last for one day."

**4. Assessment of smoke impacts**
It is unclear how the authors define air quality (AQ) observation sites as influenced or not influenced by smoke. Is smoke "influenced" defined from the perspective of the model e.g., air quality monitoring sites that were impacted by a conserved smoke in the WRFChem simulation or PM2.5 or CO levels greater than non-fire simulation? Or is smoke influenced defined by AQ observation e.g., PM2.5 > some threshold. The criteria for smoke influenced needs to be clearly defined. And the rational for the criteria explained.

There are too many figures and the accompanying discussions are difficult to follow. I feel the study would be better served had the authors focused on a handful of days using air quality sites that were smoke impacted, from the simulations' perspective using a clear, well defined definition of smoke impacted (e.g., WRF-Chem conserved smoke tracer, CO
levels, etc.)

**Response:** Thanks for pointing out the missing definition of the smoke influence. We updated the definition in the Section 3.1, Lines 368-371 of the revised manuscript: "Here we define the fire influence based on the $PM_{2.5}$ impact from fire. If the $PM_{2.5}$ difference between sim_nofire and sim_FINN is less than 1 µg m$^{-3}$ over a specific region (and time), then this region (and time) is not influenced by fire smoke. This value is near the low end of the thresholds often used to assess the smoke impacts (Munoz-Alpizar et al. 2017, Matz et al. 2020). "

Because we focused on a regional smoke impact on different sites and from different fire events, the logic of picking the studied time period was based on the strength of the fire.

5. L9: "The emissions of duff burning were estimated based on a field measurement"

**Response:** We rewrote the sentence to "The emissions of duff burning were calculated using the measured duff fuel consumption for the 2016 Rough Ridge Fire and emission factors."

6. L24-26: "Fires contribute 26.9% of total volatile organic compounds (VOC) emissions and 27.5% of PM emissions in the U.S. according to the 2014 US Environmental Protection Agency (EPA) National Emissions Inventory (NEI) (USEPA, 2017)."

**Response:** Thanks for pointing it out. We double checked the EPA NEI 2014 report and rewrote this sentence to: "Fires contributed to 13.6% of total volatile organic compounds (VOC) emissions and 27.5% of PM emissions in the U.S. according to the 2014 US Environmental Protection Agency (EPA) National Emissions Inventory (NEI) (USEPA, 2017)." This number was calculated based on the following chart (a part of figure 4 of the NEI2014 report).

[Figure]

7. L33-34: "Wildfires produce about 3.5% of global tropospheric ozone production, though ozone production rates of individual fires vary with fuel type, combustion efficiency, etc. (Alvarado et al., 2010; Jaffe and Wigder, 2012)."

Location, time of year, meteorology, and pre-existing atmospheric composition are likely important factors as well.

**Response:** We modified the sentence to: "Wildfires produce about 3.5% of global tropospheric ozone production, though ozone production rates of individual fires vary with location, time, fuel type, combustion efficiency, meteorology, and local pre-existing atmospheric composition, etc."

8. L44-45: "In many regions around the world, including the U.S., wildfires have an increasing trend during recent decades…"
Elaborate on what kind of increasing trend? Frequency of large fires, fire severity, burned area?

**Response:** We modified the sentence to: "In many regions around the world, including the U.S., wildfires have an increasing trend during recent decades, in both the number and the area of total large fires (Dennison et al., 2014; Barbero et al., 2015). In addition, weather with high fire potential has appeared more frequently. (Yang et al., 2011; Jolly et al., 2015; Abatzoglou and Williams, 2016)…"

9. L49-55: Paragraph needs rewriting. Introductory sentence of the paragraph is about human health impacts of smoke, but two of three following sentences discuss radiative impacts of smoke aerosol. I suggest dropping the radiative impact sentences and added more information on health impacts.

**Response:** We dropped the radiative impacts sentences and filled more information on the health impact, at Lines 50-56, the paragraph now becomes:

Negative impacts of wildfires on human health are devastating when smoke plumes are transported to populated metropolitan areas (Kunzli et al., 2006). Epidemiological studies have revealed fire emissions' contribution to $PM_{2.5}$ oxidative potential, which is related to respiratory and cardiovascular diseases (Verma et al., 2014; Yang et al., 2016; Fang et al., 2016). During the fire events in northwestern U.S. during August-September, a regional mortality of 183 due to $PM_{2.5}$ exposure was estimated, in which 95% was contributed by fire emissions (Zou et al., 2019). Based on the U.S. respiratory hospital admissions and additional premature deaths during and after fire events, the economic loss is $11 – 20 billion due to short term exposures, and $76 – 130 billion due to long-term exposures (Fann et al., 2018).

10. L71-72: "Duff typically represents the detritus or dead plant organic materials fallen at the top layer of soil."
This is a good location to define the terms "duff", "peat", and "organic soil". They are often used interchangeably when discussing global wildland fire. For example, a couple sentences down the authors use "organic soil".

**Response:** Responded together with the next comment.

11. L82-83: "Besides duff, peat is another burnable organic soil that typically represents the fermentation below the duff layer (Frandsen, 1987)."
See previous comment.

**Response:** We moved the sentence in L82-83 "Besides duff, peat is another burnable organic soil that typically represents the fermentation below the duff layer (Frandsen, 1987)" back to L72-73, and added another sentence after this: "'Organic soil' is often used to represent soil formed by plant and animal decomposition, including peat and duff. Duff, peat and organic soil were sometimes used interchangeably, and we focus on duff in this study."

12. L105: Define "prescribed fire"

**Response:** We changed this sentence to "On one hand, most fires in the southeastern US are prescribed (planned) and…", now at Line 107-108.

13. L106: Yokelson et al. (2013) is not a good reference for this statement, more appropriate reference(s) needed.

**Response:** We changed the reference to the following report from US Forest Service:

Waldrop, T.A. and Goodrick, S.L., 2012. Introduction to prescribed fires in Southern ecosystems. Science Update SRS-054. Asheville, NC: US Department of Agriculture Forest Service, Southern Research Station. 80 p., 54, pp.1-80.

14. L115-116: Change to: "…estimated to account for approximately 60% of total PM2.5 emitted from the fire."

**Response:** Updated (Lines 117-118).

15. L131: "…temperate forest duff emission factor of nitrogen oxides (NOx) is 0.67 g/kg…" Citation needed.

**Response:** Cited Yokelson et al. (2013) (Line 133).

16. L137-142: This last paragraph of the Introduction needs to provide a better, but still brief, overview of the study (similar to the abstract).

**Response:** We added two more sentences (L142-143): "The simulated concentrations of air pollutants were compared with those from observations, between burns with and without duff, and between PM2.5 and ozone."

17. L144: 2.1 Study Region
The authors should provide a map of the study region with polygons of fire perimeters and markers for urban areas of interest in the air quality simulations. Include scale bar. This is necessary for the reader and will allow the authors to streamline this section which reads very rough.

**Response:** The map the reviewer suggested was shown in Figure S2 in the previous manuscript. In the revision, we moved it to the main text (Figure 1) to help readers streamline the section. This

figure is referenced in Line 172 and Line 235.

18. L165: 2.2 Fire cases
Maps of each fire subregion region with fire boundary polygons should be provided. Perhaps a three panel – entire study region (see comment above), southern Appalachian region, and the Okefenokee swamp region.

**Response:** In the revision, Fig S2 was replaced as a fire subregion plot showing the fire hotspots from FINNv1.5 and the specific duff burning region we investigated for each fire case.

19. L173-185: It would be interesting if the authors could provide a couple sentences on the fire history of the Okefenokee swamp region. Three large fires in a short period, how does this compare with fire history at the swamp?

**Response:** We added the following sentences (L192-195): "Historical fire records from 19th century revealed the strong connection between drought and Okefenokee fires, leading to a 'Okefenokee drought-fire cycle' (https://www.frames.gov/catalog/34075, last access: November 5, 2021). Although the Okefenokee fires from 2007 to 2017 was more frequent than the historical mean, more information on the fire cycle change is needed."

20. L209-210: Refer reader to figure for domain. Also, I suggest swapping order of Fig S1 and S2. Present model domains first, then time series of OC emissions related to fire activity.

**Response:** As mentioned in the response to the comment 17, we moved the Figure S2 to the main text Figure 1, showing model domains, fire regions and the nearby cities.

21. L240-241: "The fire emissions from FINNv1.5 were implemented into WRF-Chem by Pfister et al. (2011a), which contains the daily burned area and emissions of an amount of gas and aerosol species with a spatial resolution of 1 km (Wiedinmyer et al., 2011)."
More explanation is needed here. Did the WRF-Chem installation that you used include FiNNv1.5 fire emissions on an hourly time step that could be included in simulations? I didn't think that was an option. Or did you download FiNNv1.5 fire emissions with MOZART speciation and ingest these emissions into WRF-Chem? If the latter, how did you convert daily emissions of FiNNv1.5 (https://www.acom.ucar.edu/Data/fire/data/README_FINNv1.5_08112014.pdf ) into hourly input for WRF-Chem?

**Response:** The WRF-Chem setting used in this study does not use daily fire emission data directly in the model. We converted FINNv1.5 data to hourly emission input using the gridding program from NCAR ACOM (https://www.acom.ucar.edu/Data/fire/). As described in the response to the comment 3, we assume the fire lasts for one day and emit evenly during the 24 hours. We added more detailed description in Lines 265-267: "No a-priori diurnal cycle of the fire emission was applied in the WRF-Chem model, and the hourly fire emission applied in the WRF-Chem simulations was the hourly emission converted from the daily fire cases from FINN assuming each observed fire hotspots last for one day."

22. L265: "The duff burning contributed 60% of the total PM2.5 emission" Should read: "The duff burning was estimated to have contributed 60% of the total PM2.5 emission"

**Response:** Changed (Line 292).

23. L291-295: The authors should note that the Geron & Hays (2013) was a field study that made in-situ measurements of EFPM2.5 from three different peat fires in coastal North Carolina. Black et al. is a laboratory study that measured EF from peat core samples from two locations in North Carolina. BTW – Black et al. (2016) is missing from bibliography.

**Response:** Thanks for pointing this out. We updated this in Lines 316-318 and updated the bibliography.

24. L303-305: Please provide reference for NOx EF or refer reader to Table 2. The authors do not mention Table 2 anywhere in the text. Table 2 should be referenced when discussing EF.

**Response:** In the revision, we refer Table 2 in Line 329. As responded to the comment 15, NOx EF is also referenced.

25. L306-314: This is a reasonable approach at the Okefenokee fire sites, to estimate duff reduction from previous burns (2007 to 2011, 2007/2011 to 2017).

**Response:** As described at Lines 341-342, "87% of the burned area in Oke11 was burned by Oke07, and 79% of Oke17 was burned by the 2007 and 2011 fires." The duff reduction from previous burns should include both flaming and smouldering, but the smouldering duff reduction is hard to estimate for limited information. We calculated the fuel load change of the late fires based on an estimated duff layer recovery rate of 1 mm/year. We added Table S4 to show the estimated burned depth change in the Oke11 and Oke17 case due to the previous burns, referred at Line 348.

26. L322-325: "We did not use the commonly used approach to scale up the FINN emissions because we wanted to understand if the missing duff burning contributed to the underestimate FINN emissions to a certain extent. This FINN emission underestimate would lead to uncertainty in quantitatively estimating the contribution relative to the above-ground fuel consumption."
This is unclear and must be rewritten.

**Response:** We removed the first sentence. This part (now Lines 356-358) was changed to "This FINN emission underestimate would lead to uncertainty in quantitatively estimating the contribution relative to the above-ground fuel consumption. To roughly assess the uncertainty, we did a sensitivity experiment by doubling FINN emissions for the Oke07 case (Exp_FINN, Table 1)."

Section 3.1
27. It is unclear how the authors define air quality (AQ) observation sites as influenced or not influenced by smoke. Is smoke "influenced" defined from the perspective of the model e.g., air

quality monitoring sites that were impacted by a conserved smoke in the WRFChem simulation or PM2.5 or CO levels greater than non-fire simulation? Or is smoke influenced defined by AQ observation e.g., PM2.5 > some threshold. The criteria for smoke influenced needs to be clearly defined. And the rational for the criteria explained.

**Response:** We updated Lines 368-371 of the revised manuscript: "Here we define the fire influence based on the $PM_{2.5}$ impact from fire. If the $PM_{2.5}$ difference between sim_nofire and sim_FINN is less than 1 $\mu g\ m^{-3}$ over a specific region (and time), then this region (and time) is not influenced by fire smoke. This value is near the low end of the thresholds often used to assess the smoke impacts (Munoz-Alpizar et al. 2017, Matz et al. 2020). " This is also mentioned in the response to the comment 4.

28.Figure S3 is not that helpful in discerning agreement between base simulation (No fire) and AQ observations. A time series like Figure 3 would be far more useful (with smoke influence clearly defined, previous comment).

**Response:** We replaced Figure S3 with the time series similar to Figure 3 bur only for the observed sites and time that are not influenced by fire smoke.

29. State how many air quality sites were used in each domain.

**Response:** We added the information on site numbers in Section 2.3.3, Line 252-255: "56 $PM_{2.5}$ observation sites and 53 ozone observation sites are included in the evaluation for App16, 76 $PM_{2.5}$ observation sites and 225 ozone observation sites for Oke07, 112 $PM_{2.5}$ observation sites and 215 ozone observation sites for Oke11 and PB11, 120 $PM_{2.5}$ observation sites and 208 ozone observation sites for Oke17, 38 $PM_{2.5}$ observation sites and 101 ozone observation sites for ER08."

30. The PM2.5 emission and transport from duff burning 30. L381-382: "Thus, implementing duff burning doubles the PM2.5 concentrations from App16"
Is this statement based on specific AQ site(s)? Please clarify.

**Response:** This sentence was updated to "Thus, implementing duff burning doubles the fire-induced PM2.5 concentrations during App16 over the study domain." (Lines 418-419)

31. L384-385: "The total burned area of Oke07 was 5 times more than that of App16. The emissions were larger from Oke07 and correspondingly the simulated PM2.5 concentrations are greater."
Since Oke07 lasted > 2 months, it would be more useful to compare area burned and emissions for the periods of the simulations, perhaps as daily average.

**Response:** This sentence was updated to: "The total burned area of Oke07 was 5 times more than that of App16, and over the periods of the simulations, the daily average fire emission during Oke07 was 3 times more than that during App16 (Figure S1). Correspondingly, the simulated PM2.5 concentrations during Oke07 are greater." (Lines 421-423)

32. L386-388: "In the sim_FINN+duff runs, the simulated fire plume effectively approaches the underestimated regions, but the enhancement is still not enough over some regions."
By approaches, I assume the authors mean the simulated surface PM2.5 in the plume approaches the concentration of the AQ observations showing greatest impact? Please clarify.

**Response:** This sentence was updated to: "In the sim_FINN+duff runs, the simulated surface PM2.5 in the fire plume effectively approaches the underestimated regions showing greatest fire impact, but the enhancement is still not enough over some regions." (Lines 424-426)

33. L408-409: "The daily variations are different between observations and simulations because the observed fire emission dataset was at daily rather than hourly intervals."
I commented on this topic earlier. How did the authors temporally distribute (daily to hourly) the fire emissions for WRF-Chem input? The temporal distribution of fire emissions is critical to getting realistic simulations.

**Response:** Thanks for pointing this out. As previously responded, we did not apply a diurnal cycle to the fire simulations. More clarification was made at L265-267: "No a-priori diurnal cycle of the fire emission was applied in the WRF-Chem model, and the hourly fire emission applied in the WRF-Chem simulations was the hourly emission converted from the daily fire cases from FINN assuming each observed fire hotspots last for one day."

Technical
It would make for a more pleasant read if for URLs the authors provided citations in the text and the links in the bibliography, e.g.:
"(http://www.gatrees.net/forest-management/forest-health/alertsand-updates/Wildfire%2 0Damage%20Assessment%20for%20the%20West%20Mims%20Fire.pdf, last access: December 3,185 2020)."

**Response:** Thanks for the suggestion. During our previous submissions for the ACP journal, this citation format agrees with the journal requirement of citing and referencing. So we kept this for the current manuscript.

L33: "Wildfires produce about…" to "Wildfires account for…"

**Response:** Changed. (Line 33)

L49: "…when fire plumes are transported…" to "…when smoke plumes are transported…"

**Response:** Changed. (Line 50)

L75-76: "…may provoke each other…" Rephrase, be more specific about the physical processes to which you are referring.

**Response:** This sentence was removed in the revision.

L83: change "swamp" to "wetland", the latter encompasses swamp, bog, march, etc.

**Response:** Changed. (Line 83 and Line 84)

L86: "…are also evaluated…" to "have also been evaluated…"

**Response:** Changed. (Line 87)

L88: "However, the air quality impacts of emissions from duff fires are very limited…" I don't think this is what the authors intend to state.

**Response:** This sentence was modified to "However, the air quality impacts of emissions from duff fires are still not well understood over some regions." (Line 89)

L109: Change "springs" to "spring"

**Response:** Changed. (Line 111)

L216: Should refer to Fig S2.

**Response:** Corrected. (Line 235)

Additional English usage / technical corrections needed at: 154, 175-177, and other places

**Response**: We appreciate the detailed suggestions for English and technical improvements. The sentence at Line 155 was changed to: "Because of the sufficient light and the regularly high humidity the duff layer is accumulated in the southeastern US and contributes as the potential below-ground fuel, especially in wildlife refuges or regions where deciduous trees are widely distributed with lack of prescribed burn removal." Lines 180-183 was changed to "The 2007 Okefenokee mega wildfire was ignited in the Okefenokee Wildlife Refuge (30.67° N, 82.45° W) on April 16, and had burned more than 500,000 acres until late June (Fire Behavior Assessment Team, 2007). Protracted drought led to low water levels in the Okefenokee swamp and provided the condition of burning in a mix of shrub scrub, wetland prairies, duff, cypress and long-leaf pine forests." Some other English improvements and typo corrections were also made.

**References**

Fann, N., Alman, B., Broome, R.A., Morgan, G.G., Johnston, F.H., Pouliot, G. and Rappold, A.G.: The health impacts and economic value of wildland fire episodes in the US: 2008–2012. Science of the total environment, 610, pp.802-809, https://doi.org/10.1016/j.scitotenv.2017.08.024, 2018.

Lin, S., Sun, P., and Huang, X.: Can peat soil support a flaming wildfire? International Journal of

Wildland Fire, 28(8), 601-613, https://doi.org/10.1071/WF19018, 2019.

Carlyn J. Matz, Marika Egyed, Guoliang Xi, Jacinthe Racine, Radenko Pavlovic, Robyn Rittmaster, Sarah B. Henderson, David M. Stieb, 2020, Health impact analysis of PM2.5 from wildfire smoke in Canada (2013–2015, 2017–2018). Science of The Total Environment, 725 (10), 138506

Matz, C.J., Egyed, M., Xi, G., Racine, J., Pavlovic, R., Rittmaster, R., Henderson, S.B. and Stieb, D.M., 2020. Health impact analysis of PM2. 5 from wildfire smoke in Canada (2013–2015, 2017–2018). Science of the Total Environment, 725, p.138506.

Munoz-Alpizar, R., Pavlovic, R., Moran, M.D., Chen, J., Gravel, S., Henderson, S.B., Menard, S., Racine, J., Duhamel, A., Gilbert, S., Beaulieu, P.A., Landry, H., Davignon, D., Cousineau, S., Bouchet, V., 2017. Multi-year (2013–2016) PM2.5 wildfire pollution exposure over North America as determined from operational air quality forecasts. Atmosphere 8, 179.

Prichard, S.J., O'Neill, S.M., Eagle, P., Andreu, A.G., Drye, B., Dubowy, J., Urbanski, S. and Strand, T.M., 2020. Wildland fire emission factors in North America: synthesis of existing data, measurement needs and management applications. International Journal of Wildland Fire, 29(2), pp.132-147. https://doi.org/10.1071/WF19066

Zou, Y., O'Neill, S.M., Larkin, N.K., Alvarado, E.C., Solomon, R., Mass, C., Liu, Y., Odman, M.T., Shen, H.: Machine Learning-Based Integration of High-Resolution Wildfire Smoke Simulations and Observations for Regional Health Impact Assessment. Int. J. Environ. Res. Public Health, 16, 2137. https://doi.org/10.3390/ijerph16122137, 2019

**Responses to Comments from Referee #2**

The authors are addressing a much-needed topic in fire emission calculations and air quality modeling. When ground fuels burn they become a large source of trace gases and aerosols into the atmosphere and there is a great lack of data needed to quantify these emissions. This manuscript highlights the need for more information, and how the lack in the current available information hinders air quality analyses. Below I have specific comments designed to help make this analysis more robust.

Response: Thank you for reviewing our manuscript and providing very constructive and in-depth comments and suggestions.

**Specific Comments**

**Fire Events:** Four of the largest wildfire events in the southeastern US were selected for air quality modeling with WRF-Chem for two domains. One domain simulates fires in the southern Appalachian mountains and one domain simulates fires in southern Georgia. The fires ranged from 91K acres to > 500K acres. One thing I found lacking was consideration of the Evans Road and Pains Bay wildfires. They were smaller than the top 4 criteria (41K, and 5K acres respectively), but were significant in terms of the emissions from the burning of organic/ground fuels and subsequent air quality impacts. Rappold et al. 2011 conducted a health impact analysis from the Evans Road wildfire and Tinling et al. 2016 conducted a similar analysis for the Pains Bay wildfire. The first two sentences of the Abstract state that "Wildfires can significantly impact air quality and human health. However, little is known about how duff and peat burning contributes to these impacts." Given the goals of this paper, I would expect these studies/impacts be part of the introduction and also a consideration in this study.

**Response:** Thank you for this valuable comment. We paid attentions only to fire size when were selecting fire cases for this study but ignored some important fires that occurred in the sites with rich organic soils despite relatively smaller size. As suggested, we conducted additional simulations of the 2008 Evans Road fire and the 2011 Pains Bay fire. The corresponding modifications to the manuscript are follows.

(1) Study region (Lines 166-169): We added a third area of coastal eastern North Carolina.
(2) Fire cases (Lines 197-204): We added simulation cases for the two fires in Table 1 with description in section 2.2. The 2008 Evans Road fire ignited on June 3, 2008 and burned 41060 Acres. The 2011 Pains Bay fire ignited on May 5 2011 and burned 29400 Acres. The simulation periods for the two fires were June 2008 and May 2011, respectively. A domain centered in North Carolina was used.
(3) Results: Two figures were added to illustrate the simulation results of the two fires. Figure 12 shows results from a typical fire day during the 2008 Evans Road fire (June 12, 2008), including the simulation-observation comparison in the Sim_nofire, Sim_FINN, and Sim_FINN_duff cases, the $PM_{2.5}$ enhancement due to duff burning, and time series from

June 7 to June 15 in an urban site and a rural site close to the burning region. Figure 13 is the corresponding results during the 2011 Pains Bay fire, including a typical fire day (May 12, 2011) analysis and the time series from May 6 to May 15. The evaluation of ozone is not shown because, similar to the 4 fire cases discussed in the manuscript, the ozone effect from duff is weak in comparison with the above-ground fuel effects. The results indicate that the PM2.5 enhancement from duff flaming is significant in both fire cases. The smoke transport effects more to the cities by the 2008 Evans Road fire and less during the 2011 Pains Bay fire, because the 2011 Pains Bay fire occurs under prevailing wind to the ocean.

(4) Discussion (Lines 465-471): We discussed the two cases in the revised manuscript,

**Duff Flaming Phase Emissions:** Why only focus on flaming emissions of duff? Smoldering phase emissions are important in terms of air quality impacts (as noted by Rappold et al. 2011 and Tinling et al. 2016). Smoldering of ground fuels does not always occur on a time scale of months to year, it can occur on the scale of hours/days, and while the plumes do not necessarily loft high, during the day they mix near the surface (where people breath) under the mixing height and can be transported further distances. At night they can transport along terrain features often impacting small towns in rural areas closer to the fires. This becomes an environmental justice issue as well. Limiting the work here to only flaming phase duff emissions unnecessarily limits the utility of this study.

**Response:** We agree with this point about the importance of smoldering smoke. Besides the two studies mentioned in the comment, recent studies such as Kim et al. (2018) and Chan et al. (2020) also emphasized the health effects due to wildfire smoldering. We did not investigate smoldering smoke for two reasons. First, smoldering smoke is mainly a local process. Most smoldering cases only impact towns and rural regions close to a burning site. This study focused on the regional air quality impacts of smoke, especially in remote large cities with large populations. Secondly, WRF-chem which we used is a regional model. The local smoke process during smoldering phase is not well described in a regional model. We are planning to dig into the duff smoldering phase more in a separate study using different modeling tools such as the PB-P specific local smoke model (Liu et al. 2018). We added more discussion in the revised manuscript (Line 617-620).

**Duff Consumption:** Related to this is how much duff consumption actually went into each of the scenarios? I see that 4.6 cm of duff burned in the 2016 Rough Ridge fire which went into the App16 case (fuel loading 3.15 kg/m2). How many centimeters of duff burned in the Oke07, Oke11 and Oke17 cases? And was the same fuel loading from App16 (3.15 kg/m2) assumed for the Oke scenarios? The end of section 2.4.2 discusses how regrowth was handled, but again, what actual data went into the scenarios? I recommend adding the duff depth burned and fuel loading estimates to a table. Further, were all duff estimates assumed to burn in the flaming phase? Or were some estimated to burn in the smoldering phase (and thus eliminated if I am interpreting the discussion regarding the focus on flaming phase emissions)?

**Response:** The only measured data of duff burning used in this study was from the App16 case. Because of this limitation, we assume the 4.6 cm per day duff flaming rate for all the studied cases in the Southeastern US except Oke11 and Oke17. Duff at some grid points was already burned by

the fire prior to each of the two fires. We used a simple algorithm to estimate the depth of duff layer at the grid points. The lack of duff measurements in Oke sites and the two added sites is one of the uncertainties for this study, which is discussed in the manuscript. We assessed the uncertainty by conducting the sensitivity test which changes the duff flaming emission by ± 20% (discussed in Section 3.4). We clarified this in the revised manuscript (Lines 362-364) and added the duff depth burned and fuel loading in Table S4.

Yes, we only considered flaming burning in this study. The 4.6 cm duff burning are all from flaming phase. The carbon emission during the prolonged duff smoldering is larger in magnitude than PM2.5 emissions but with little regional air quality impact.

**Duff Consumption:** Section 2.4.2 indicates that "we estimated duff emissions and added them to FINN." Was a full suite of trace gas and aerosol species added? Or were only PM2. 5, NO and NO2 species added to the simulations? At a minimum I would expect a full suite of species using default above-ground fuel emission factors would be added to represent the duff fuels, and ideally those emission factors be adjusted based on available literature for duff fuels. Recent studies for the SE in George et al. 2016 and Black et al. 2016 may be useful. They both conducted lab experiments based on peat from North Carolina. Many of these trace gases have implications for ozone and secondary aerosol formation.

**Response:** Thanks for pointing it out. Not only $NO_x$ and $PM_{2.5}$, we also had added a full suite of trace gas and aerosol species for duff emissions when we conducting the simulations. We included the duff emission of other species in the revised manuscript. In Table S2, we added the gas species duff burning emission factors used in this study. The duff emission species we added to the simulations are: CO, NO, $NO_2$, $SO_2$, $NH_3$, NMOC, BIGALK, BIGENE, $C_{10}H_{16}$, $C_2H_4$, $C_2H_5OH$, $C_2H_6$, $C_3H_6$, $C_3H_8$, $CH_2O$, $CH_3CHO$, $CH_3COCH_3$, $CH_3COOH$, $CH_3OH$, ISOP, MEK, MVK, TOLUENE, $PM_{2.5}$ (OC and BC). This list is based on the species list from the above-ground emissions in FINN v1.5. For the VOC species, the emission factors are from the organic soil burning emission factors summarized in Yokelson et al. (2013).

**VOC-limited:** Section 3.3. Is the SE (App16 domain) really in a VOC-limited scenario in the winter (Nov)? I recommend showing estimates to support this.

**Response:** The following estimations are used to confirm the VOC-limited scenario in the November of 2016. First, we show a relationship between the EPA measurements of daytime (8 am - 7 pm) Ozone and $NO_2$ concentrations in Georgia and North Carolina, a negative correlation is shown in the below figure:

[Figure]

Then, in the response to the next issue, we made a sensitivity run doubling the duff burning $NO_x$ emissions, and during the App16 case, increasing duff $NO_x$ emissions further decreases ozone (Fig S26 in the revision), which also supports that November is in a VOC-limited scenario over the southeastern US. Some previous studies also mentioned that the ozone scenario is turning from $NO_x$-limited to VOC-limited from fall to winter in the eastern and southeastern US (Jacob et al., 1995; Simon et al., 2015; Zhang et al., 2016).

**NOX and Ozone Generation:** Much of the discussion focuses on how the duff did not add much ozone to the model simulations, which is attributed to the NO/NO2 emission factors being low. There needs to be more discussion about the variability in NO/NO2/NOX emissions from duff. Yokelson et al. 2013 is just a single experiment and Urbanski 2014 applies an uncertainty of 100% to the data. Studies such as Burling et al. 2010, McMeeking et al. 2009, Selimovic et al. 2018 and Clements and McMahon 1980 are all studies that measured emission factors for ground fuels. NO values range from 0.56 to 2 g/kg, and NO2 values range from 0.23 to 2.7 g/kg. These data argue for perhaps using greater EF values for NO and NO2 and also (especially) sensitivity runs that vary the NOX EF's by more than just 20%. Recommendation: I recommend an additional sensitivity run using 100% per Urbanski 2014 (e.g. 2x duff).

**Response:** One of the conclusions of this work is that while the PM effect of duff is as significant as that of the above-ground fuel, the ozone effect from duff is not as strong as the effect from the above-ground fuel. For example, Burling et al. (2010) showed that the emission factor from duff is 0.738 g/kg for NO, and 0.232 g/kg for $NO_2$, and the emission factor from the above ground fuel is $1.720 \pm 0.454$ g/kg for NO and $1.023 \pm 0.286$ g/kg for $NO_2$ (Table 3 in Burling et al., (2010)). The summary of the $NO_x$ emission factors mentioned in the reviewer's comment is included in Table S3. Corresponding description is added in the revised manuscript (Lines 525-531).

To address the reviewer's concern of the uncertainty due to the $NO_x$ emission factor, we made sensitivity runs for the App16 and Oke07 cases using twice $NO_x$ from duff burning (the '2x duff

NOx' case). As shown in Fig. S26, doubling duff $NO_x$ further decreased ozone during the App16 case. During the Oke07 case, ozone increases corresponding with the increased $NO_x$ emissions from duff, but the ozone effect is still weaker than the PM effect. The corresponding description is added to Section 3.4 (Lines 525-531).

**Meteorology/Transport:** Section 3.2 discusses the PM2.5 emissions and transport. I recommend making the discussion more robust by including references to support the statement "Both biases in fire emission calculation and smoke transport simulation should
be the contributors." I recommend Li et al. 2020 and Garcia-Menendez et al. 2013.

**Response:** Thanks for the suggestion. We cited Li et al. 2020 and Garcia-Menendez et al. 2013 in the revised manuscript.

**Technical Corrections** Table 1 caption needs to include more information such that the information in the table is understandable independent of the paper. Also, I recommend adding fire size (acres) to the table as well.

**Response:** As suggested, the title of Table 1 is revised to "The simulation period and fire emission inventories applied in different WRF-Chem simulations and experiments." Fire size information and the information about the new 2x_duff_NOx case is added in the table.

**References used in the responses:**

Jacob, D.J., Horowitz, L.W., Munger, J.W., Heikes, B.G., Dickerson, R.R., Artz, R.S. and Keene, W.C., 1995. Seasonal transition from NOx-to hydrocarbon-limited conditions for ozone production over the eastern United States in September. Journal of Geophysical Research: Atmospheres, 100(D5), pp.9315-9324.

Liu Y-Q, Goodrick S., Achtemeier G, 2018, The Weather Conditions for Desired Smoke Plumes at a FASMEE Burn Site. Atmosphere 2018, 9(7), 259; https://doi.org/10.3390/atmos9070259

Simon, H., Reff, A., Wells, B., Xing, J. and Frank, N., 2015. Ozone trends across the United States over a period of decreasing NOx and VOC emissions. Environmental science & technology, 49(1), pp.186-195.

Zhang, Y. and Wang, Y., 2016. Climate-driven ground-level ozone extreme in the fall over the Southeast United States. Proceedings of the National Academy of Sciences, 113(36), pp.10025-10030.